# Molecular basis of XPF-ERCC1 targeting to SLX4-dependent DNA repair pathways

Junjie Feng [1], Peter R. Martin [2], Szymon Kowalski[2,3], Maxime Lecot [2,5], Nora B. Cronin [4], Teige Matthews-Palmer [1], Wojciech Niedzwiedz [2] & Basil J. Greber [1] ✉

The preservation and faithful propagation of genetic information is essential for all life forms and depends on cellular pathways that enable replication, recombination, and repair of DNA. The multifunctional XPF-ERCC1 DNA endonuclease complex acts in several DNA repair pathways and interacts with numerous partner proteins and large DNA repair assemblies, including the nucleotide excision repair machinery and the SMX tri-endonuclease complex. Here, we report structures of XPF-ERCC1 in complex with the DNA repair factors SLX4 and SLX4IP, thereby identifying key residues responsible for direct interactions with XPF-ERCC1. When introduced into human cells, point mutations in these interfaces impair the interactions between XPF-ERCC1 and SLX4 or SLX4IP, and disruption of the XPF-SLX4IP interface leads to cis-platin sensitivity. Furthermore, our data reveal the structure of the human XPF-ERCC1-SLX4IP-SLX4[330-555] complex with DNA bound at its active site, and they complete the structural characterisation of molecular interactions required to assemble the SMX complex.

Maintenance of genome integrity is critical for all cellular life. Therefore, cells have evolved pathways that remove or mitigate DNA damage and facilitate interconversion of unstable DNA structures that arise during cellular processes such as DNA replication or recombination. Nucleolytic enzymes called structure-specific or structure-selective endonucleases are instrumental in several DNA repair pathways[1]. The sites of action of structure-specific endonucleases are determined by structural features of the DNA, most typically junctions between single-stranded and double-stranded DNA (ssDNA and dsDNA, respectively), rather than by DNA target sequences[1]. Due to this property, the activity of structure-specific endonucleases needs to be tightly controlled in time and space to avoid unwanted DNA incisions that can have disastrous consequences for the cell[2], such as chromosome pulverisation caused by premature activation of the SLX1-SLX4-MUS81-EME1 endonuclease complex[3]. Conversely, failure

to properly target and activate structure-specific endonucleases can impair important cellular processes, including DNA repair, DNA replication, and DNA segregation. XPF-ERCC1 (xeroderma pigmentosum complementation group F and excision repair cross-complementing group 1) is a structure-specific endonuclease complex that specifically cleaves 3'-flaps, i.e. junctions between a DNA duplex and a single-stranded 3'-overhang, and damage-containing DNA bubbles on the 5'-side of the lesion[4,5]. Mutations affecting the complex are causative of human disease, including Fanconi anaemia[6,7] and the cancer-prone UV-sensitive syndrome xeroderma pigmentosum from which the name of the catalytic XPF subunit is derived[4,8,9]. Concurrently, inhibition of XPF-ERCC1 has been suggested as a possible mechanism for sensitising cancer cells to cis-platin treatment[10].

The enzymatic activity of XPF-ERCC1 is employed in several DNA repair and genome maintenance processes, including nucleotide

[1]Division of Structural Biology, The Institute of Cancer Research, Chester Beatty Laboratories, London, UK. [2]Division of Cell and Molecular Biology, The Institute of Cancer Research, Chester Beatty Laboratories, London, UK. [3]Department of Pharmacology, Faculty of Medicine, Medical University of Gdańsk, Gdańsk, Poland. [4]London Consortium for High Resolution Cryo-EM, The Francis Crick Institute, London, UK. [5]Present address: Faculty of Medicine of Rennes, University of Rennes, F-35043 Rennes, France and Molecular Oncology, Institut Curie, PSL Research University, CNRS, UMR144, Paris, France. ✉e-mail: basil.greber@icr.ac.uk

excision repair (NER)[2], inter-strand crosslink (ICL) repair[6,7,11], homologous recombination[12], alternative lengthening of telomeres (ALT)[13], and TRF2-mediated telomere shortening[14–16]. The recruitment of XPF-ERCC1 to its DNA target sites in these pathways depends on interactions with other DNA repair factors that provide, albeit sometimes indirectly, a target-recognition function and act to enhance XPF-ERCC1 endonuclease activity. Among these XPF-ERCC1 recruiting DNA repair factors is SLX4[17], a 1834 aa-long scaffold protein that additionally binds the endonuclease SLX1 and the endonuclease complex MUS81-EME1[18–23]. This complex is called SMX (SLX-MUS-XPF) in its assembled form[18]. The activities of the participating endonucleases are enhanced by complex formation[18,21,24,25], and subsets of the three nucleases are coordinated to enable their functions in several DNA repair pathways. Specifically, XPF-ERCC1 acts in an SLX4-dependent manner to unhook inter-strand cross-links in ICL repair[26–29] and to release stalled replication forks for homologous recombination[30]. SLX1-SLX4 and MUS81-EME1 form a Holliday junction resolvase[18–22] that is further activated by a non-enzymatic contribution from XPF-ERCC1[18], and MUS81-EME1 within SMX cleaves replication forks[18], contributing to maintenance of common fragile sites. The molecular details of the SLX4-SLX1 and SLX4-MUS81-EME1 interactions have been elucidated by X-ray crystallographic and nuclear magnetic resonance (NMR) structures (refs. 31–34 and the unpublished X-ray crystal structure PDB ID 7BU5). In contrast, structural insight into the XPF-ERCC1 recruiting activity of SLX4 has remained elusive. Biochemical

and cell biological characterisation revealed that SLX4 residues 529-550 (refs. 27,35,36) residing within an SLX4 domain near the MUS312/MEI9 interaction-like region (MLR)[11,20,22] are required for the interaction (Fig. 1a).

In addition to formation of the SMX complex, SLX4 and its partner endonucleases interact with numerous additional DNA repair factors that enable the complex or its components to act in additional DNA repair and genome maintenance pathways. One of these is SLX4IP (C20orf94)[22], a protein involved in the ALT and ICL repair pathways[27,37]. In ALT, a telomerase-independent strategy used by certain cancer cells to achieve replicative immortality, SLX4IP appears to balance the activity of the Bloom syndrome helicase (BLM) and SMX by repressing BLM[37]. Within SMX-related complexes, SLX4IP has been suggested to interact with SLX4, with XPF, or with both XPF and SLX4[22,27,37]. However, the molecular basis of these interactions and the mechanism of action of the complex in ALT and ICL repair have not been elucidated in mechanistic detail. Delineating the specific function of individual players in these pathways has been hampered by the lack of structural information on key complexes, which precludes the design of separation-of-function mutations to probe the contributions of XPF-ERCC1 to individual pathways.

To investigate the interplay of XPF-ERCC1-binding DNA repair factors, we have determined cryogenic electron microscopy (cryo-EM) structures of XPF-ERCC1-XPA, XPF-ERCC1-SLX4IP, and XPF-ERCC1-SLX4IP-SLX4, followed by biochemical analysis. Our structures reveal

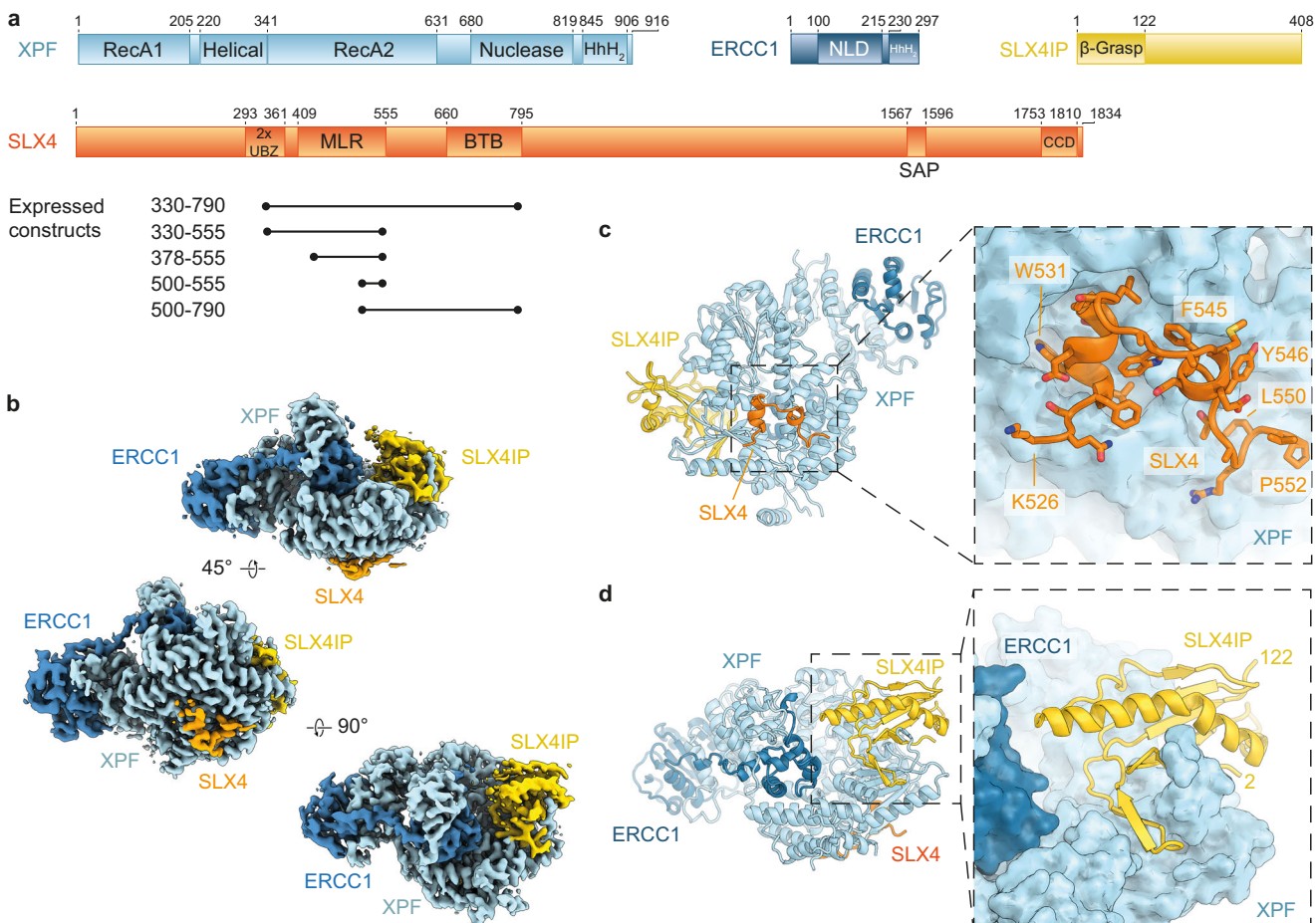

**Fig. 1 | Cryo-EM structure of the XPF-ERCC1-SLX4IP-SLX4^330-555 complex.**
**a** Schematic representation of the domain organisation of XPF, ERCC1, SLX4IP, and SLX4 (domain boundaries according to[17,24] or structural results from this work). SLX4 variants expressed for structural and biochemical experiments are indicated. Abbreviations: HhH helix-hairpin-helix, NLD nuclease-like domain, UBZ ubiquitin-binding zinc finger type 4, MLR MUS312-MEI9 interaction-like region, BTB broad-complex, tramtrack and bric-a-brac, SAP SAF-AB, acinus and PIAS, CCD conserved C-terminal domain. **b** Cryo-EM reconstruction of the XPF-ERCC1-SLX4IP-SLX4^330-555 complex. **c** Structure of SLX4 residues 526-552 bound to XPF. **d** Structure of SLX4IP bound to XPF.

the binding interfaces between XPF-ERCC1 and its partners SLX4IP and SLX4. They demonstrate that SLX4IP directly interacts with XPF without a requirement for SLX4 or other additional protein subunits, and they reveal the molecular details of the SLX4-dependent recruitment of XPF-ERCC1 to SMX. We subsequently probe these interfaces using point mutations, identifying key residues involved in both interactions. Furthermore, our structure of XPF-ERCC1-SLX4IP-SLX4-DNA reveals how DNA is accommodated at the active site of human XPF-ERCC1.

## Results

### AlphaFold2-based modelling of XPF-ERCC1 complexes

Previous studies have shown that XPF-ERCC1 is recruited to the SMX complex via an interaction with SLX4. This interaction remains incompletely characterised but is known to reside in the SLX4 region surrounding the MLR motif in the N-terminal half of the protein (residues 409–555; Fig. 1a)[11,20,22] and to encompass SLX4 residues between L530 and L550[27,35,36]. To gain insight into the recruitment of XPF-ERCC1 to the SMX complex, we generated AlphaFold2[38,39] predictions of the XPF-ERCC1-SLX4IP-SLX4 complex (Supplementary Fig. 1a, b).

The predicted structure of the XPF-ERCC1-SLX4IP-SLX4 complex shows SLX4IP and a segment of SLX4 comprising residues 330-555 (SLX4$^{330-555}$) bound to XPF (Supplementary Fig. 1a, b). In addition to several stretches of amino acids that could serve to anchor SLX4 to XPF, this prediction also suggests an interaction between the second Ubiquitin-binding zinc finger (UBZ-2) domain of SLX4 (residues 332-377) and the tandem helix-hairpin-helix (HhH$_2$) domain of ERCC1 (Supplementary Fig. 1a, b). This interaction would be incompatible with the resting state of the enzyme (Supplementary Fig. 1c–e), suggesting a possible mechanism for the previously described role of SLX4 in XPF-ERCC1 activation[18,21,24,25]. We therefore decided to test and refine these predictions by experimental structure determination of the predicted complexes.

### 2.9 Å-resolution cryo-EM structure of XPF-ERCC1

We recombinantly expressed XPF-ERCC1-SLX4IP, XPF-ERCC1-SLX4$^{330-555}$, XPF-ERCC1-SLX4IP-SLX4$^{330-555}$, XPF-ERCC1-SLX4IP-SLX4$^{378-555}$, XPF-ERCC1-SLX4IP-SLX4$^{300-790}$, XPF-ERCC1-SLX4IP-SLX4$^{500-790}$, XPF-ERCC1-SLX4IP-SLX4$^{500-555}$ (Fig. 1a), and, for comparative purposes (see below), XPF-ERCC1-XPA complexes in insect cells and purified them to homogeneity (Supplementary Fig. 2a–i). We found co-purification of all expressed subunits as envisaged, highlighting the utility of AlphaFold2 modelling results for construct design. Subsequently, we collected cryo-EM data on the purified XPF-ERCC1-XPA, XPF-SLX4IP-ERCC1-SLX4$^{330-555}$, and XPF-ERCC1-SLX4IP complexes as well as a DNA-bound XPF-ERCC1-SLX4IP-SLX4$^{330-555}$ complex and processed these data using cryoSPARC[40] and RELION[41].

The cryo-EM structure of XPF-ERCC1-XPA at 2.9 Å resolution (Supplementary Fig. 3a–e) reproduces the binding of residues 71-77 of the NER factor XPA in a groove on the surface of ERCC1 (Supplementary Fig. 3f–h) observed previously in a hybrid NMR-X-ray structure[42]. The overall architecture of XPF-ERCC1 in this cryo-EM reconstruction is in good agreement with the previously reported structure of XPF-ERCC1 in the apo-state[24]. XPF in our structure also assumes the auto-inhibited conformation with a contact between the helical and nuclease domains (Supplementary Fig. 3i)[24]. However, due to its higher resolution compared to the previously published structure, our cryo-EM map enabled us to build and refine a model of XPF-ERCC1 with improved accuracy, which we use as a basis for comparison with subsequent structures of XPF-ERCC1 bound to other DNA repair factors.

### Cryo-EM of the XPF-ERCC1-SLX4IP-SLX4 complex reveals the SLX4-XPF binding interface

We next analysed the XPF-SLX4IP-ERCC1 and XPF-ERCC1-SLX4IP-SLX4$^{330-555}$ complexes. The cryo-EM structure of XPF-ERCC1-SLX4IP-

SLX4$^{330-555}$ at 3.2 Å resolution reveals simultaneous binding of both SLX4IP and SLX4 to XPF (Fig. 1b–d and Supplementary Fig. 4a, b). A short peptide of SLX4 is visualised on one side of XPF, while the N-terminal half of SLX4IP occupies the opposite side of XPF (Fig. 1b–d and see below). Our structure shows that the N-terminal 122 residues of SLX4IP form a compact β-grasp fold, a mixed α+β-fold that occurs in a diverse array of both enzymes and structural proteins, including ubiquitin[43]. In SLX4IP, five β-strands wrap around a single long α-helix in a β$_2$-α-β$_3$ configuration. The SLX4IP structure resembles the structure of MAJIN, a component of the meiotic telomere complex, which tethers telomere ends to the nuclear envelope during meiosis (Supplementary Fig. 4c-e)[44,45]. The density attributable to SLX4 in our cryo-EM structure of the XPF-ERCC1-SLX4IP-SLX4$^{330-555}$ complex is shorter than predicted by AlphaFold2. The most clearly resolved portion encompasses only residues 526–552 (Fig. 1b, c). Additional weak density that is visible in the predicted location of residues 498–511 is not clear enough to model this segment. However, results from a DNA-bound complex suggest that it corresponds to the region of SLX4 predicted by AlphaFold2 (see below). The SLX4 UBZ-2 domain and its predicted interaction with the ERCC1 HhH$_2$ domain (Supplementary Fig. 1a, b) are not visualised, and XPF assumes the auto-inhibited conformation (Fig. 1b).

### Analysis of the XPF-SLX4 binding interface

The SLX4 segment 529–535 folds into a short α-helix that forms multiple interactions with XPF, including insertion of W531 into a hydrophobic pocket (Figs. 1c, 2a and Supplementary Fig. 4f, g). Residues 536–552 form a mostly extended segment with only minimal secondary structure (residues 542–545 assume α-helical conformation) that interacts with XPF through several side-chain contacts (Fig. 2a). It is worth noting that residues L530, F545, Y546, and L550, mutation of which was previously found to impair the SLX4-XPF-ERCC1 interaction in yeast two-hybrid experiments and cause mitomycin C sensitivity in human cells[35], are all located within the most clearly resolved, and thus likely most stably bound, SLX4 segment in our cryo-EM map (Fig. 2a). Our structure thus provides the mechanistic explanation of these observations.

To verify the functional importance of the stably bound SLX4 segment identified in our cryo-EM reconstruction under controlled conditions, we used insect cells to co-express XPF-ERCC1 and SLX4$^{330-555\ WT}$ or SLX4$^{330-555\ 5A}$, with the latter construct harbouring the L530A, W531A, L536A, Y546A, and L550A mutations in the XPF-binding motif of SLX4. Small-scale purification of the resulting complexes from insect cell lysate using a Strep tag on the SLX4$^{330-555}$ fragment showed that the SLX4-XPF interaction was disrupted by the mutations in SLX4$^{330-555\ 5A}$ (Fig. 2b). In further support of our structural findings, SLX4 residues 526-555 fused to MBP were able to form a stable interaction with XPF-ERCC1, while the interaction was lost in the MBP-SLX4$^{526-555\ 5A}$ construct containing the five alanine mutations described above (Fig. 2c). We then expressed full-length SLX4 harbouring these point mutations (eGFP-SLX4$^{5A}$) in human HEK293TN cells and performed co-immunoprecipitation (co-IP) experiments to verify our structural and in vitro biochemical results in a cellular context. Co-IP experiments from cells transfected with constructs expressing eGFP-SLX4$^{WT}$ confirmed the known interactions with the components of the XPF-ERCC1 and MUS81-EME1 endonuclease complexes (Fig. 2d). Identical experiments performed using cells transfected with the SLX4$^{5A}$ mutant construct showed a strongly impaired interaction with XPF, ERCC1, and SLX4IP, while the interactions with MUS81-EME1 remained intact (Fig. 2d).

In summary, our biochemical and cellular interaction assays support the functional predictions of our structural findings and suggest that SLX4 residues 526–552 are primarily responsible for targeting XPF to SLX4-containing complexes. Prior studies found that the Fanconi anaemia patient mutation XPF L230P and C236R, as well as the

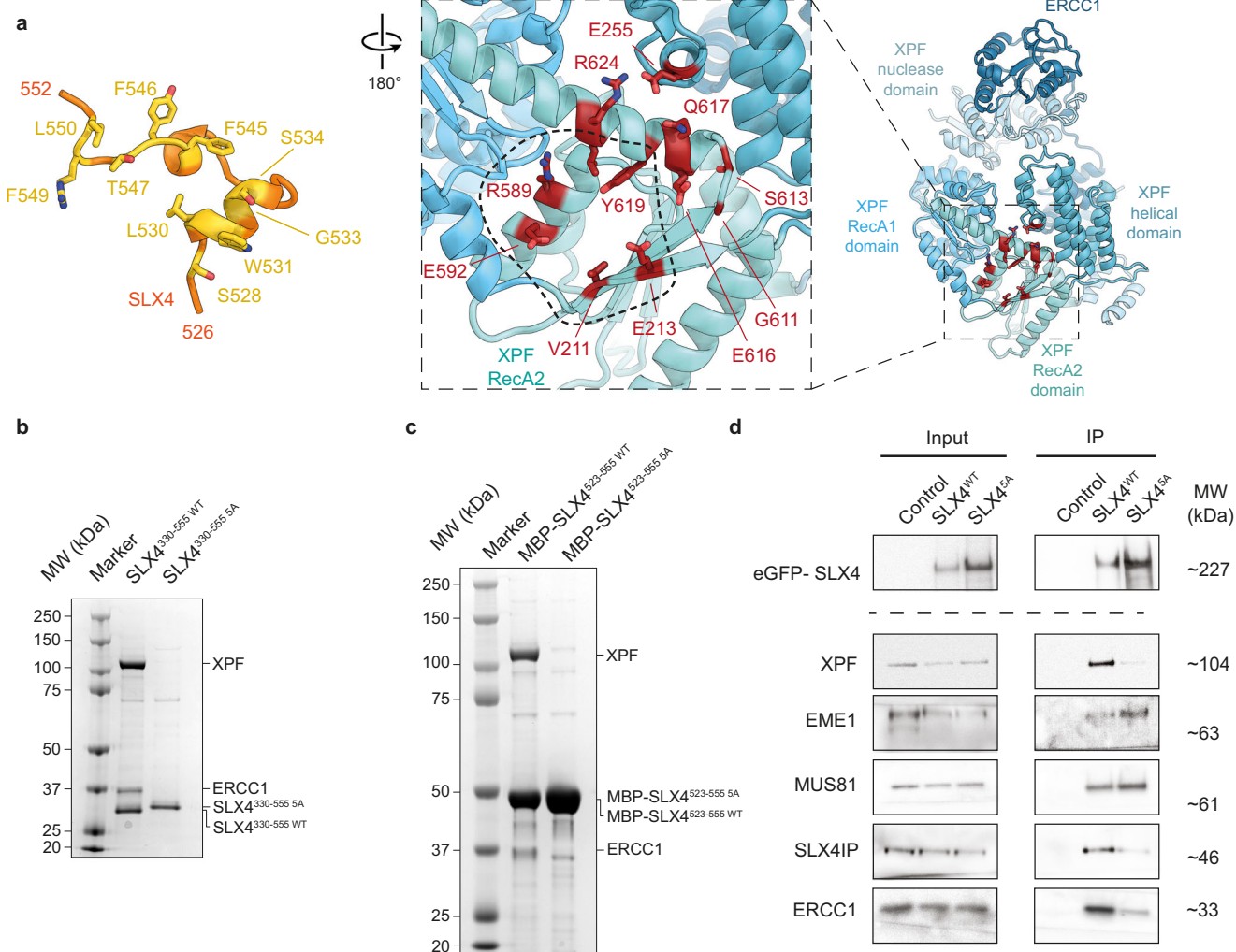

**Fig. 2 | Characterisation of the SLX4-XPF interaction. a** Structures of the SLX4 fragment 526-552, with residues interacting with XPF shown as yellow sticks (left), and structure of XPF, with residues interacting with SLX4 shown as red sticks (right). The area of XPF interacting with the SLX4 helix 529-534 is marked by a dashed line. **b** Small-scale in-vitro co-sedimentation assay of XPF-ERCC1 with Strep-tagged SLX4[330-555] wild type and 5 A mutant. **c** Small-scale in-vitro co-sedimentation assay of MBP-fused SLX4[523-555] wild type and 5 A mutant. **d** Co-IP of wild type and 5A-mutant SLX4 from HEK293TN cells. 1% of input lysate to the IP was analysed as input. Source data are provided as a Source data file.

synthetically generated mutations XPF G325E and XPF Δ323−326, show impaired interactions with SLX4, manifesting as recruitment or activation defects[23–25]. Our structure shows that these residues do not directly form part of the core XPF-SLX4 binding interface but are located adjacent to it (Supplementary Fig. 4h). This observation suggests that these mutations may induce local structural distortions in the structure of XPF that affect interactions with SLX4.

## Structure of a DNA-bound XPF-ERCC1-SLX4IP-SLX4[330-555] complex

To obtain further insight into the XPF-activating role of SLX4 (refs. 18,21,24,25. and see above), we tested the enzymatic activity of XPF-ERCC1 in complex with SLX4[330-555]. We found enhanced cleavage of a Y-fork substrate in the presence of SLX4[330-555] in vitro (Fig. 3a−c and Supplementary Fig. 5). However, all SLX4 residues observed in our XPF-ERCC1-SLX4IP-SLX4 structure are far removed from the XPF active site or any putative regulatory elements. This suggests that residues in SLX4[330-555] that were not observed in the structure might be able to transiently or dynamically form interactions with XPF-ERCC1 to increase substrate cleavage. We note that our SLX4[330-555] construct does not contain the SLX4 BTB domain, which was hypothesised to contribute to XPF activation previously[25]. Possible candidates for such

stimulatory interactions thus include, but are not limited to, the predicted interaction between the UBZ-2 domain and the ERCC1 (HhH)$_2$ domain, which would preclude formation of the auto-inhibited conformation of XPF-ERCC1 (Supplementary Fig. 1c−e).

To test whether the interactions of XPF-ERCC1 with SLX4 might change in the presence of a bound DNA substrate, we prepared an XPF-ERCC1-SLX4IP-SLX4[330-555]-DNA complex containing a Y-fork DNA substrate with phosphorothioate modifications of the backbone around the preferred cleavage site of XPF (Fig. 3d). The cryo-EM structure of this complex at 3.4 Å resolution (Fig. 4a, Supplementary Fig. 6a, b) exhibits the conformational change of the XPF-ERCC1 HhH$_2$ domain dimer observed in a previously reported XPF-ERCC1-DNA structure (Supplementary Fig. 6c-e)[24], and a dsDNA stem is seen sandwiched between the relocated HhH$_2$ domains and the remainder of XPF, without forming any interactions with SLX4IP (Fig. 4a, b). The features of the DNA in our cryo-EM map indicate the presence of B-form DNA on both sides of the active site, which is unexpected, given the 3'-flap endonuclease activity of XPF-ERCC1 and the previous low-resolution analysis of a MUS81-EME1 complex with a 3'-flap DNA substrate[46] (Supplementary Fig. 6e, f). We modelled the sequence register of the DNA according to the best match between the known sequence of the supplied DNA substrate and the purine-

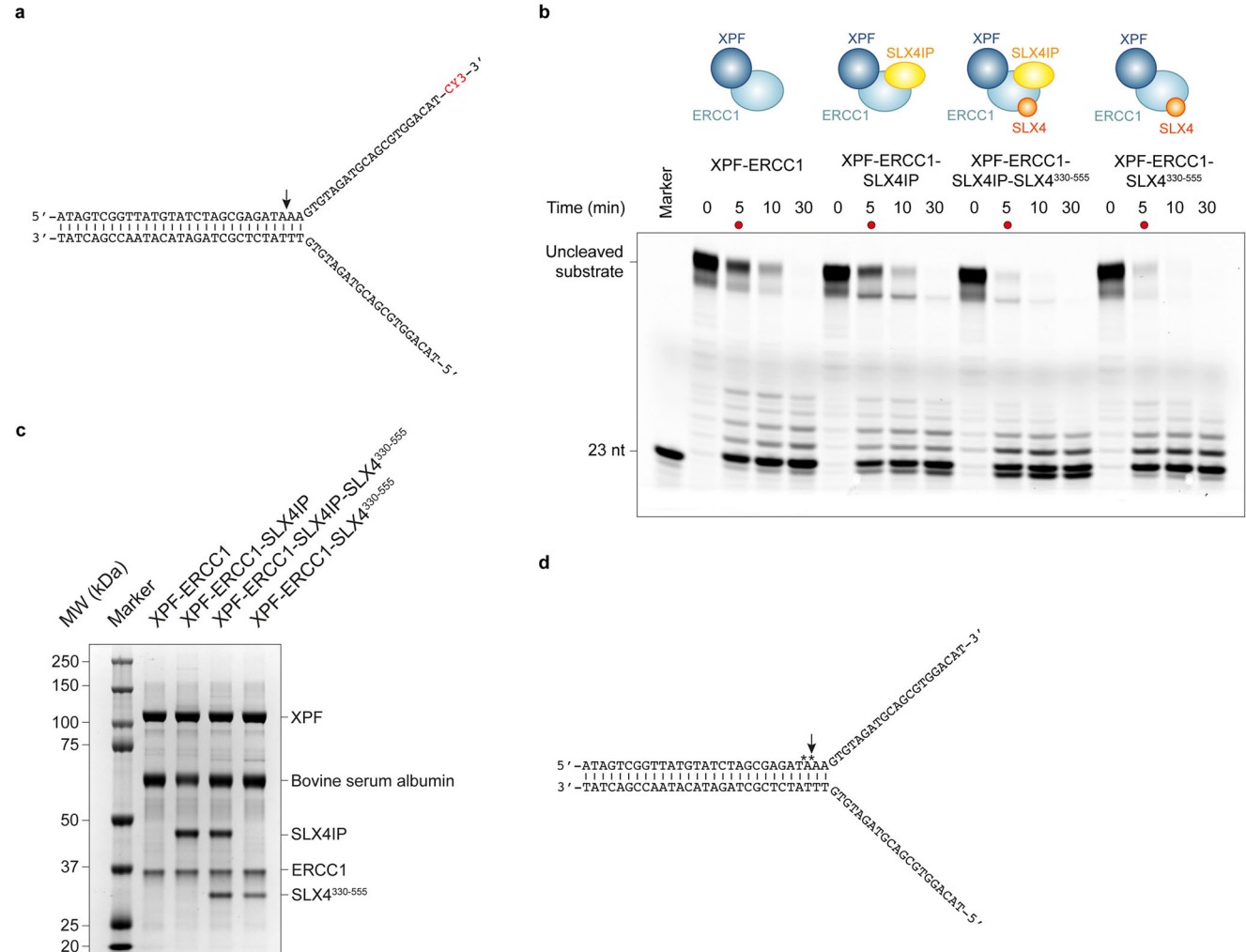

**Fig. 3 | XPF-ERCC1 endonuclease activity in the presence of SLX4IP and SLX4^330-555.** **a** Y-fork DNA substrate used to assay activity of XPF-ERCC1 complexes. The preferred cleavage site of XPF-ERCC1 is denoted by an arrow and liberates a 23-nt DNA fragment with a 3'-Cy3 fluorophore used for detection of the product. **b** Comparison of nuclease activity of XPF-ERCC1, XPF-ERCC1-SLX4IP, XPF-ERCC1-SLX4IP-SLX4^330-555, and XPF-ERCC1-SLX4^330-555. Conversion of uncleaved input substrate (a) into product (23-nt fragment) was monitored by detection of Cy3 fluorescence. SLX4IP has no impact on activity, while the presence of SLX4 in the complex increases cleavage, as evidenced by the more rapid disappearance of the band corresponding to the intact substrate. To facilitate visualisation, the 5 min time points are marked with red dots. **c** Coomassie stained SDS-PAGE gel of protein sample processing control (before final 10x dilution into the nuclease reaction) to confirm that equivalent amounts of endonuclease were present in all samples. Two additional repeats of the experiment shown in panels b and c are provided in Supplementary Fig. 5. **d** Y-fork DNA substrate used to assemble an XPF-ERCC1-SLX4IP-SLX4^330-555-DNA complex for cryo-EM. Bonds protected against endonucleolytic cleavage by phosphorothioate linkages are indicated by asterisks (*). Source Data are provided as a Source data file.

pyrimidine pattern in the cryo-EM density (Supplementary Fig. 6g). Even though the purine-pyrimidine pattern establishes the prevalent sequence register of the bound DNA substrate, we acknowledge that, given the absence of ssDNA-dsDNA junction recognition, a small fraction of complexes might contain the DNA bound in a different sequence register. This analysis suggests that the enzyme exclusively bound the dsDNA stem of the substrate, with the XPF helical domain acting as a "stopper" for the base-paired end of the dsDNA. In this configuration, the ssDNA-dsDNA junction is located in the solvent, beyond the XPF-ERCC1 HhH2 domain dimer and far away from the active site. The reason for the failure of XPF-ERCC1 to recognise the ssDNA-dsDNA junction in our substrate is unclear. However, consistent with the orientation of the DNA in our structure, dsDNA nicking activity has been observed previously for a murine mini-SLX4-XPF-ERCC1 complex[29]. The comparably low efficiency of this dsDNA nicking reaction might be why we were able to observe uncleaved DNA substrate in our cryo-EM complexes.

## Architecture of the DNA-bound XPF active site

In agreement with the previously reported human XPF-ERCC1-DNA structure[24], XPF-ERCC1 undergo conformational changes upon DNA binding. Most prominently, the HhH2 domains move by approximately 45 Å, and the structural module comprising the N-terminal domain of ERCC1 and nuclease domain of XPF rotates such that the contact with the auto-inhibition loop within the XPF helical domain breaks[24] (Supplementary Fig. 6c–e). However, possibly due to the absence of crosslinking reagents in our sample, the positioning of the HhH2 domains after this structural transition is different in our structure (Supplementary Fig. 6h, i), allowing the DNA in our complex to form more extensive interactions with XPF-ERCC1 and to directly interact with the XPF active site (Fig. 4c–e). The DNA is bound by interactions with the ERCC1 HhH2 domain, the XPF nuclease domain, and the XPF RecA1 domain (Fig. 4c–e). Proximity of the DNA to residues in the XPF helical domain is likely due to the double-stranded nature of the segment that XPF-ERCC1 bound to in our in vitro reconstituted system, and these interactions may not meaningfully contribute to binding of

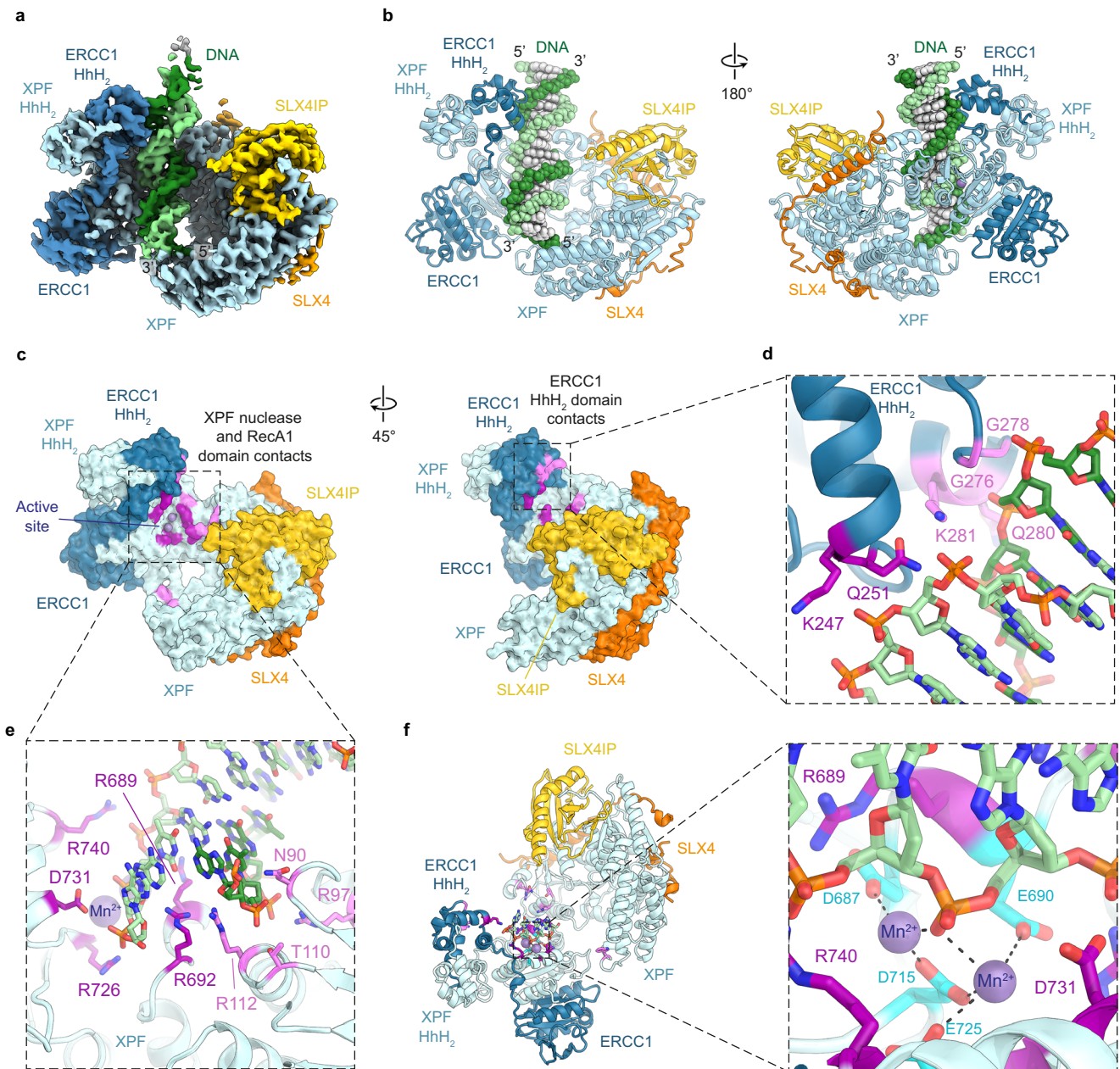

**Fig. 4 | Structure of a DNA-bound XPF-ERCC1-SLX4IP-SLX4[330-555] complex.**
**a** Cryo-EM map of the XPF-ERCC1-SLX4IP-SLX4[330-555]-DNA complex. XPF is shown in cyan, ERCC1 in blue, SLX4 in orange, SLX4IP in yellow, the scissile DNA strand in light green, and the non-scissile strand in dark green. **b** Atomic model of the XPF-ERCC1-SLX4IP-SLX4[330-555]-DNA complex (colours as in **b**). **c** Surface view of the proteins in the complex with residues in proximity to DNA (<3.5 Å) shown in pink. DNA is not shown. **d** Close-up view of the DNA contacts with the ERCC1 HhH$_2$ domain. **e** Close-up view of the DNA contacts with the XPF nuclease and RecA1 domains. Manganese ions bound in the active site shown as spheres. **f** View of the XPF active site with two metal ions coordinated by negatively charged residues (shown in cyan) and a DNA backbone phosphate.

cellular DNA substrates. However, given the relative arrangement of double-stranded DNA and the XPF helical domain (Fig. 4a, b), it is possible that the helical domain prevents XPF-ERCC1 from binding to continuous double-stranded DNA or serves as a wedge that prevents re-annealing of forked substrates, e.g. in the context of replication fork cleavage.

The interactions at the ERCC1 HhH$_2$ domain are primarily formed between the phosphodiester backbone of the non-scissile strand of the DNA and backbone amides of G276, G278, Q280, and K281 and the side chain amide of Q280 (Fig. 4d). Backbone and side-chain atoms of K247 and Q251 may additionally contribute to interactions with the scissile strand, though these interactions are less well defined (Fig. 4d). At the XPF nuclease and RecA1 domains, multiple arginine side chains (R97,

R112, R689, R692, R726, R740) insert into the DNA minor groove or interact with the phosphodiester backbone of either DNA strand (Fig. 4e). R689A and R726A mutations in this group of side chains have been found to strongly impair XPF-ERCC1 activity[47] (residue numbering in this reference is offset by 11 residues), and R689 is mutated in Fanconi anaemia[48].

In the active site, four negatively charged side chains (D687, E690, D715, E725) are found in direct proximity to additional density that cannot be attributed to protein or DNA residues. One peak of this density coincides with the location of magnesium ions observed in X-ray crystal structures of Aeropyrum pernix XPF, the Pyrococcus furiosus MUS81/XPF homolog Hef, and human MUS81 (Supplementary Fig. 6j–m)[46,49,50], confirming the presence of a catalytic metal ion in a

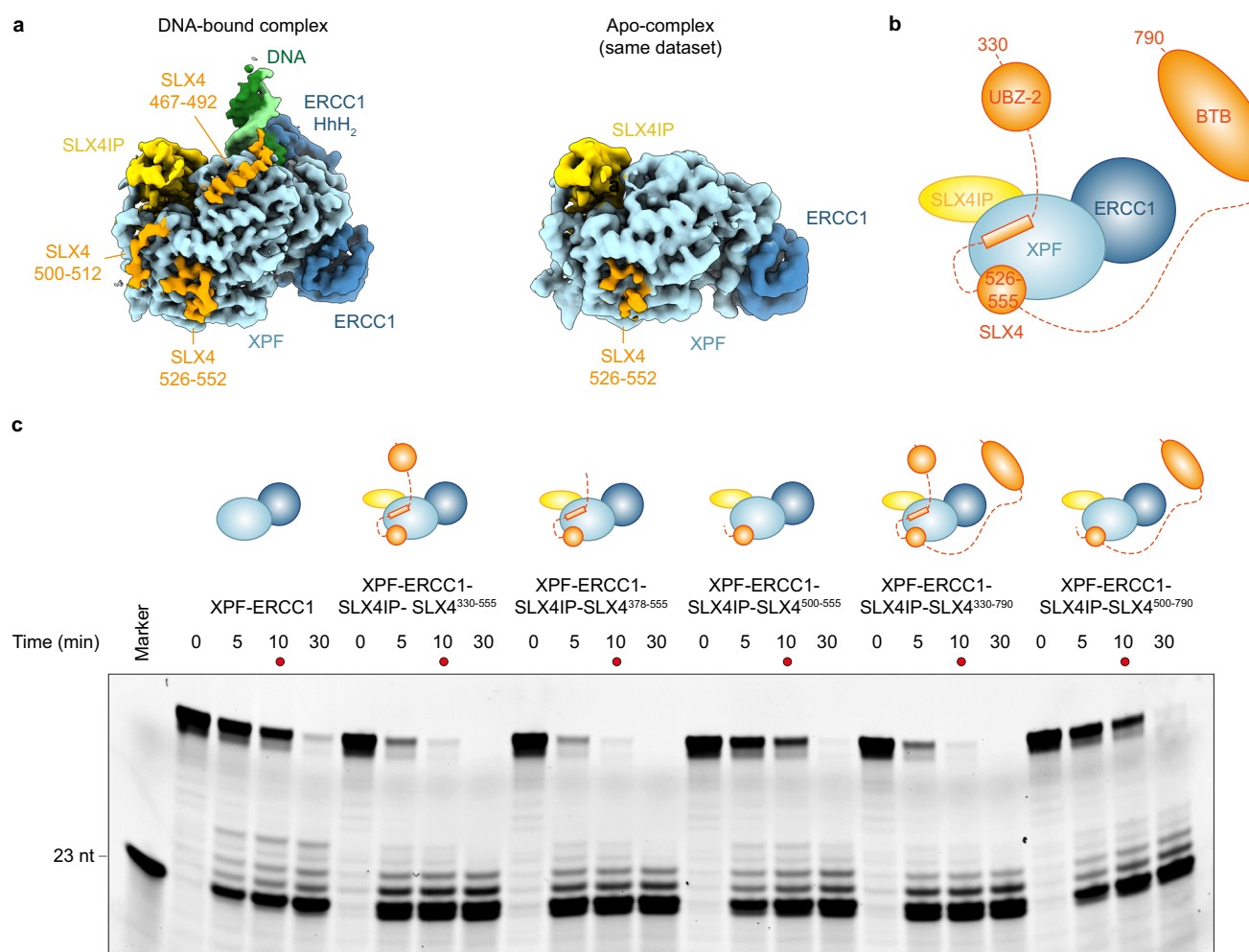

**Fig. 5 | Additional SLX4 segments resolved in the structure of the XPF-ERCC1-SLX4IP-SLX4$^{330\text{-}555}$-DNA complex. a** Left: Cryo-EM map of the XPF-ERCC1-SLX4IP-SLX4$^{330\text{-}555}$-DNA complex. Three SLX4 segments (orange) are visualised and labelled. Right: Cryo-EM map of the DNA-free complex from the same data collection shown in the same orientation. Only SLX4 residues 526-552 are visualised, in line with the higher-resolution reconstructions shown in Fig. 1. Maps are shown without any b-factor sharpening applied to facilitate visualisation of weaker densities. **b** Schematic depiction of the sequence elements and domains contained in the SLX4 constructs used for in vitro endonuclease assays of XPF-ERCC1-SLX4IP-SLX4 complex variants. **c** In-vitro endonuclease assay comparing the activity of XPF-ERCC1 to the activities of different variants of the XPF-ERCC1-SLX4IP-SLX4$^{330\text{-}555}$ complex. Conversion of the input substrate (see Fig. 3a) into 23-nt product was monitored by detection of Cy3 fluorescence. To facilitate visualisation, 10 min time points are marked with a red dot. The removal of the UBZ-2 domain and the addition of the BTB domain do not have an appreciable effect on endonuclease activity. Removal of the sequence elements that are solely visualised in the DNA-bound XPF-ERCC1-SLX4IP-SLX4$^{330\text{-}555}$ complex strongly reduces cleavage activity. Two additional experimental repeats and all protein sample processing controls are provided in Supplementary Fig. 7a–e. Source Data are provided as a Source data file.

conserved binding site. However, given its size and shape, the density can accommodate a second metal ion (Fig. 4f and Supplementary Fig. 6j, n). This is consistent with a prior prediction based on the enzymatic properties of XPF[51] and the prevalence of negatively charged residues in the vicinity of both putative metal binding sites. Mutation of D687, which is in proximity of the second XPF metal binding site (Fig. 4f), has been found to strongly impair XPF-ERCC1 activity[47], consistent with a role in catalysis. Alternatively, the density we attribute to a second metal ion could be a water molecule. However, observation of water molecules is unexpected for a cryo-EM map at 3.4 Å resolution, rendering this interpretation less likely. While the X-ray crystal structures of XPF, MUS81, and Hef show single metal ions bound and the respective active sites[46,49,50], some more distantly related members of the type II restriction endonuclease family, to which all of these enzymes belong[49], contain two magnesium ions bound in their active site[52] (Supplementary Fig. 6o). Overall, the available data are consistent with the presence of a second metal ion in our cryo-EM

structure and with the idea that XPF is an enzyme employing two-metal-ion catalysis.

## XPF-ERCC1-activating SLX4 interactions in the DNA-bound XPF-ERCC1 complex

Compared to the apo-structure of XPF-ERCC1-SLX4IP-SLX4$^{330\text{-}555}$, the DNA-bound complex reveals additional parts of SLX4 bound to XPF. Specifically, an α-helix formed by SLX4 residues 476–492, as predicted by AlphaFold2, is clearly visualised in this complex, along with SLX4 residues 467–475 positioned N-terminally of this helix (Fig. 5a). Weak connecting density along the predicted path of residues 493–512 is visible (Fig. 5a). It is worth noting that this segment is not visualised in the DNA-free complex obtained from a different particle subset from the same grid (Fig. 5a and Supplementary Fig. 6p), indicating the possible allosteric effects between DNA and SLX4 binding to XPF-ERCC1. This observation also corroborates our assessment that SLX4 residues 526-555 are primarily responsible for SLX4 recruitment to

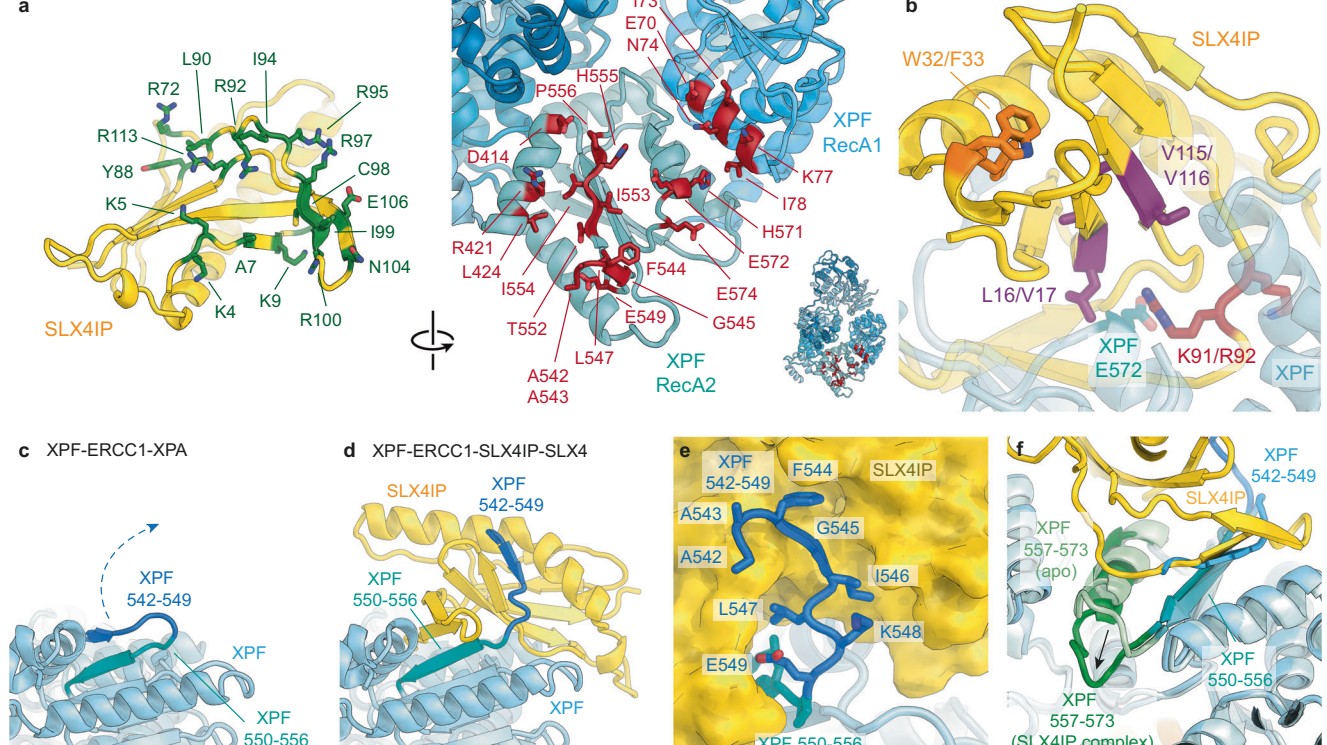

**Fig. 6 | Analysis of the XPF-SLX4IP interaction. a** SLX4IP and XPF residues mediating the SLX4IP-XPF interaction are coloured green (left) and red (right), respectively. **b** Mapping of mutations that have been reported to disrupt the SLX4IP-XPF interaction. **c** Conformation of XPF in the absence of SLX4IP.

**d** Conformation of XPF after strand exchange involving XPF residues 542–556 upon SLX4IP binding. **e** Interactions mediated by XPF residues 542–549. **f** Conformational change in XPF residues 557–573 upon binding of SLX4IP (light green: free XPF; dark green: SLX4IP-bound XPF).

XPF-ERCC1, while additional SLX4 segments can bind dynamically or in a state-specific manner.

To test the functional implications of these observations, we performed in vitro cleavage assays using XPF-ERCC1-SLX4IP-SLX4 complexes containing SLX4 segments of different lengths (Figs. 1a and 5b, c and Supplementary Fig. 7a–e). We found that SLX4 is still able to stimulate cleavage activity even after removal of the UBZ-2 domain from our SLX4 construct (resulting in SLX4$^{378-555}$). This biochemical result is in line with the fact that the SLX4 UBZ-2 domain is not visualised in our DNA-bound structure (Fig. 5a and Supplementary Fig. 7f), even though its predicted binding site on the ERCC1 HhH$_2$ domain is accessible in this DNA-bound complex (Supplementary Fig. 7g, h). We therefore conclude that the SLX4 UBZ-2 domain does not contribute to XPF-ERCC1 activity under the conditions tested. Further truncation of SLX4 to produce SLX4$^{500-555}$, thereby removing the SLX4 α-helix (residues 476-492) that is seen docked onto XPF in the DNA bound complex, reduces SLX4-mediated XPF-ERCC1 stimulation in our in vitro cleavage assay substantially (Fig. 5c and Supplementary Fig. 7d, e). This is consistent with a role of the SLX4 MLR in the stimulation of XPF-ERCC1 activity, in addition to its well-documented role as a primary SLX4-XPF interaction site[11,20,22,25,27,35,36].

Another region of SLX4 that has been implicated in enhancing XPF-ERCC1 activity is the BTB domain, which has been suggested to aid in positioning of the nuclease on its substrates by formation of transient interactions[23,25]. Extension of SLX4$^{330-555}$ and SLX4$^{500-555}$ to include the BTB domain (resulting in SLX4$^{330-790}$ and SLX4$^{500-790}$, respectively) did not have an impact on the cleavage activity of XPF-ERCC1 in our in vitro system (Fig. 5c and Supplementary Fig. 7d, e).

These data suggest that the primary, direct XPF-ERCC1-stimulatory activity of SLX4 resides in the region adjacent to the minimal XPF-binding module. This includes the SLX4 α-helix comprising residues 476–492, which becomes ordered in the DNA-bound

XPF-ERCC1-SLX4IP-SLX4$^{330-555}$ complex. Additional interactions, possibly involving the SLX4 UBZ-2 or BTB domains, were not captured in our cryo-EM experiments or enzymatic assays, but might occur in the context of larger macromolecular assemblies, different substrates, or depending on post-translational modifications.

## SLX4IP interactions with XPF-ERCC1

Our cryo-EM map of the XPF-ERCC1-SLX4IP-SLX4$^{330-555}$ complex shows that the N-terminal domain of SLX4IP interacts primarily with the RecA-like domains of XPF, with the RecA2 domain providing most of the interacting residues (Fig. 6a). This is in agreement with prior work that identified the XPF helicase-like domain as an important contributor to SLX4IP interactions[27]. The C-terminal half of SLX4IP is disordered and not visualised in the cryo-EM map. In contrast to previous hypothesis[22,27], our structure does not show any interaction between SLX4 and SLX4IP, indicating that SLX4IP can bind to XPF-ERCC1 in the absence of SLX4. We confirmed this hypothesis by purification of XPF-ERCC1-SLX4 and XPF-ERCC1-SLX4IP complexes (Supplementary Fig. 2b, c). We further characterised the binding of SLX4IP to XPF-ERCC1 on its own by determination of the structure of an XPF-ERCC1-SLX4IP complex, which resulted in a nearly identical 3.2 Å-resolution cryo-EM reconstruction, except for lacking the density attributed to SLX4 (Supplementary Fig. 8a–c). Our data and interpretation are thus consistent with reports of a direct XPF-SLX4IP interaction[37] and provide an explanation for previous data showing that an SLX4 variant carrying point mutants in the region between residues L530 and L550 was impaired in its interactions with both XPF[35] and SLX4IP[27]. The latter observation was hypothesised to indicate that SLX4 residues 530–550 might be engaging in mutually exclusive interactions with both SLX4IP and XPF[17]. However, taken together, the data are more readily explained by XPF interacting with SLX4 directly and bridging the SLX4-SLX4IP interaction, thereby accounting for the loss of the association

between SLX4 and SLX4IP when the XPF-SLX4 interaction is lost. While our data establish that SLX4 and SLX4IP can bind to XPF-ERCC1 on their own, we cannot exclude that their binding might be synergistic, i.e., that the presence of one binding partner might enhance binding of the other. Our data also do not exclude the formation of direct interactions between SLX4IP and other parts of SLX4 that are not involved in XPF-ERCC1 binding, for example, within the N-terminal 200 residues of SLX4[37].

SLX4IP residue pairs L16/V17 and V115/V116 have been identified as putative SUMO-interacting motifs (SIMs) with a possible functional role in telomeric localisation of SLX4IP[27,37]. In our structures of XPF-bound SLX4IP, these residues are buried as part of the interaction surface with XPF and are inaccessible to other proteins (Fig. 6b and Supplementary Fig. 8d–f). Our structure is consistent with the observed disruption of the XPF-SLX4IP interaction when those residues were mutated to lysines[27,37]. The L/V to K amino acid substitutions likely interfere with the XPF interaction directly, either by steric hindrance caused by the long lysine side chain or by interfering with formation of a buried hydrophobic interface. Any role in telomeric localisation of these residues may therefore have to occur in the free SLX4IP protein or in complex with other DNA repair factors such as BLM[37], which might leave these residues accessible.

The mutation of two further SLX4IP residue pairs has been found to disrupt the SLX4IP-XPF interaction: K91A/R92A and W32A/F33A[27]. Our structure shows that K91 and R92 are located on a loop approaching the surface of XPF and that the mutation of R92 disrupts a salt bridge with XPF E572 near the centre of the SLX4IP-XPF interaction surface (Fig. 6a, b and Supplementary Fig. 8f). In contrast, SLX4IP W32 and F33 are far from any visualised interaction interface (Fig. 6b and Supplementary Fig. 8d, e, g), suggesting that these mutations might act by compromising the native three-dimensional structure of SLX4IP.

### SLX4IP binding induces conformational changes in XPF-ERCC1

The conformation of XPF differs between the XPF-ERCC1-XPA and XPF-ERCC1-SLX4IP-SLX4[330-555] structures. The former structure exhibits an XPF conformation that is very close to the structure of the isolated XPF-ERCC1 complex[24]; by contrast, the presence of SLX4IP induces two conformational rearrangements in XPF. First, SLX4IP binding releases the XPF β-strand 542-547 from its position in the β-sheet in the XPF RecA2 domain (Fig. 6c). In its place, SLX4IP residues 96-100 occupy the edge of the XPF RecA2 domain β-sheet (Fig. 6d and Supplementary Fig. 8h, i). The displaced XPF residues 542-547 in turn form a β-strand addition with SLX4IP residues 7-10 (Fig. 6d, e and Supplementary Fig. 8h, i). These molecular contacts suggest a very tight interaction, in agreement with co-purification of the proteins across several purification steps (Supplementary Fig. 2c, d, Methods). The β-strand-mediated interactions of SLX4IP with XPF reveal a striking resemblance with the MAJIN-TERB2 interaction in the meiotic telomere complex (Supplementary Fig. 4c–e)[44,45]. This indicates that SLX4IP and MAJIN are structural homologs, and that their ability to form protein-protein interactions by β-strand addition may be a shared feature of this subgroup of the β-grasp protein family.

A second SLX4IP-induced conformational change occurs in XPF residues 558-573 (Fig. 6f). XPF residue A565 in the conformation observed in the absence of SLX4IP would be incompatible with the presence of SLX4IP due to steric hindrance and therefore moves out of the way upon SLX4IP binding, along with flanking residues. XPF residues 558-573 show the most pronounced positional shifts, and changes throughout the remainder of the XPF RecA1, RecA2, and helical domains are very small (Cα displacement < 1 Å).

### Disruption of the SLX4IP-XPF interface by mutagenesis

To further probe the XPF-SLX4IP interaction interface experimentally, we investigated a set of 4 XPF mutants with side chain substitutions in this interaction interface. These included the XPF I554P and E572R point mutations, the XPF F544A, I546A, L547A, E549K quadruple mutation (544-549 3AK), and a combination of all these residue substitutions (Fig. 7a). Upon purification of these mutant complexes from insect cells, we found a similarly diminished interaction between XPF and SLX4IP in all mutants (Supplementary Fig. 9a). Our experiments also confirmed that the interaction with ERCC1 (Supplementary Fig. 9a) and the in vitro enzymatic activity of the mutants against a Y-fork substrate (Supplementary Fig. 9b) remained intact, indicating that these mutations leave the folding of XPF unaffected. Overall, these findings suggest that both the SLX4IP-XPF β-strand addition interaction (targeted by the I554P mutation) and the SLX4IP interaction with the exchanged XPF β-strand (Fig. 6d, e; targeted by the quadruple 544–549 3AK mutation) are important for the recruitment of SLX4IP to XPF. To confirm these findings under physiological conditions, we performed co-IP experiments from HEK293TN cell lysates and found that the 544–549 3AK mutation in the domain-swapped β-strand of XPF strongly diminished co-purification of SLX4IP (Fig. 7b and Supplementary Fig. 9c).

Previous work has established a role for SLX4IP in ICL repair[27] and ALT[37]. However, these studies employed knock-out or mutagenesis of SLX4IP, precluding clear conclusions as to whether the observed phenotypes were due to ablation of critical interactions with XPF-ERCC1, or due to defects arising from the absence of fully functional SLX4IP. Having constructed XPF mutations that selectively disrupt the XPF-SLX4IP interaction while preserving the availability of fully functional cellular SLX4IP enabled us to selectively probe the functional role of the interaction of SLX4IP with XPF-ERCC1. We generated XPF knock-out cell lines using CRISPR-Cas9 and re-complemented these cells using wild type (XPF[WT]) and mutant (XPF[544-549 3AK]) variants of XPF using Flp-In technology. Compared to cells expressing XPF[WT], cells complemented with XPF[544-549 3AK] showed markedly reduced survival in a cis-platin survival assay despite similar XPF expression levels (Fig. 7c and Supplementary Fig. 9d). This indicates that the SLX4IP-XPF interaction itself, rather than just the availability of intact SLX4IP protein, is important for survival of cis-platin induced lesions.

## Discussion

Formation of the SMX complex relies on the recruitment of the XPF-ERCC1, MUS81-EME1, and SLX1 endonucleases by the scaffolding subunit SLX4. Previous studies indicate that being embedded within the SMX complex exerts regulatory effects on its component endonucleases[18,21,24,25], and that the recruitment and activation of XPF-ERCC1 in the context of the SMX complex may rely on distinct sets of interactions[25]. The structures reported in our study and our biochemical assays show that SLX4 residues 526-555 are necessary and sufficient for formation of a stable complex between XPF-ERCC1 and SLX4. Besides this stable interaction module, additional SLX4 regions, observed in our structure of a DNA-bound complex, predicted by AlphaFold2, or identified in previous studies, are likely to interact with XPF-ERCC1 in a transient or state-specific fashion to perform regulatory roles. Indeed, in vitro endonuclease assays suggested enhanced XPF-ERCC1 activity within our SLX4[330-555]-containing complexes. We found that truncating SLX4 to residues 500-555, thereby removing an SLX4 α-helix (residues 476-492) that is flexible in XPF-ERCC1-SLX4IP-SLX4[330-555] but contacts XPF in the context of a DNA-bound complex, strongly reduced this stimulation of XPF-ERCC1 activity. Because this part of SLX4 does not directly interact with the substrate, the XPF active site, or sections of the complex undergoing large structural transitions upon activation, an allosteric mechanism appears to be the most likely explanation for its stimulatory effect. Thus, our findings are in general agreement with the idea that full XPF activation requires formation of a distinct set of contacts compared to complex formation as such. Given the homology between XPF and MUS81, it will be interesting to dissect the similarities and differences

**a**

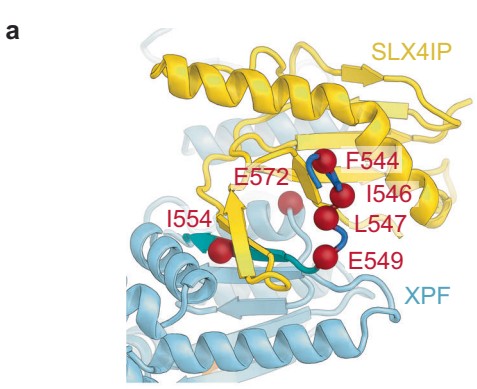

**b**

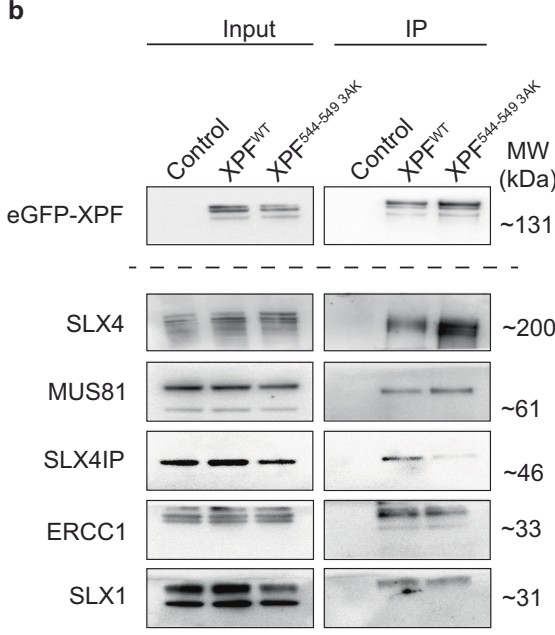

**c**

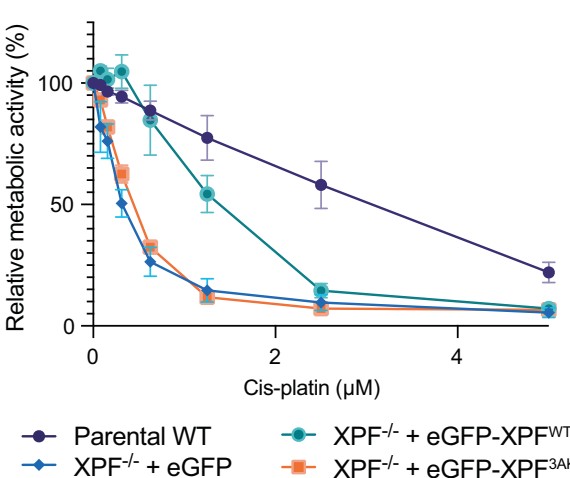

**Fig. 7 | Biochemical analysis of the XPF-SLX4IP interface. a** Locations of XPF mutations analysed (red spheres) on the structure. The 544-549 3AK mutant comprises the F544A, I546A, L547A, and E549K mutations in the domain-swapped β-strand of XPF. I554 and E572 were mutated individually and in combination with other mutations. **b** IP of wild-type and 544−549 3AK mutant eGFP-XPF from HEK293TN cells. 1% of input lysate to the IP was analysed as input. **c** Cis-platin survival assay using XPF knock-out cells complemented with eGFP-XPF[WT] and eGFP-XPF[3AK] mutant protein along with the parental strain and eGFP controls. Data from n = 3 biological repeats, each comprising 6 technical replicates, are displayed (as mean ± standard error of the mean). Relative metabolic activity in the absence of cis-platin was set to 100% for each cell line. Source Data are provided as a Source data file.

interface we observe in our cryo-EM structure does not include any residues bearing post-translational modifications, explaining why XPF-ERCC1 are constitutive subunits of SLX4-associated endonuclease complexes. This is in contrast to MUS81-EME1, which forms high-affinity interactions upon phosphorylation of MUS81 and the SLX4 MUS-binding region (MBR) by casein kinase 2 and cyclin-dependent kinase 1 in the $G_2$/M phase of the cell cycle[3,19,31,53]. A second difference between the incorporation of XPF-ERCC1 and MUS81-EME1 into SLX4-bound complexes is that recruitment of MUS81-EME1 appears to be coupled to activation while XPF-ERCC1 can bind to SLX4 via interactions that are distant to any of the likely regulatory sites within XPF, preserving its auto-inhibited conformation.

In conclusion, our results reveal the molecular basis of XPF-ERCC1 recruitment to SLX4-dependent DNA repair pathways, and they visualise the structure and XPF-binding interface of SLX4IP. Due to the involvement of SLX4IP in replicative immortality of certain cancers[54], knowledge of its structure and interactions may facilitate the discovery of next-generation small molecule or peptide therapeutics. Additionally, we have obtained the high-resolution structure of DNA-bound human XPF-ERCC1. This structure visualises the activated conformation of XPF-ERCC1 in the context of the XPF-ERCC1-SLX4IP-SLX4[330-555] complex and provides detailed insight into molecular contacts between XPF and its substrate near the active site. Thereby, our analysis provides a structural framework for the future mechanistic dissection of the functions of the SMX complex and associated proteins in DNA repair and maintenance of genome integrity.

## Methods
### Cloning of protein expression constructs
**Source of DNA sequences.** The coding sequence for XPF (HsCD00295463, deposited by the ORFeome Collaboration) was obtained from PlasmID[55]; sequences encoding SLX4 (HsCD00821809) and SLX4IP (HsCD00784820) were obtained from DNASU (deposited by the Center for Personalized Diagnostics)[56]; the codon-optimised sequence encoding ERCC1 and XPA was obtained by gene synthesis (Twist Biosciences). A list of synthetic DNA sequences created during this study is provided in Supplementary Data 1.

### Basic expression constructs
DNA sequences encoding recombinant proteins used in this study were amplified from the donor plasmids using polymerase chain reaction (PCR) and cloned into pFastBac 438-series vectors suitable for protein expression in insect cells[57,58] using InFusion cloning kits (Takara)[59]. The DNA sequences encoding XPF and XPA were amplified by PCR and cloned into the Ssp1 site of the 438-B vector, which harbours an N-terminal His$_6$ tag, producing 438-B-XPF and 438-B-XPA. The sequences encoding ERCC1 and SLX4IP were cloned into the Ssp1 site of the 438-A vector, expressing the proteins without a tag, producing 438-A-ERCC1 and 438-A-SLX4IP. Sequences encoding SLX4[330-555], SLX4[330-555 5A], SLX4[378-555], SLX4[500-555], SLX4[330-790] and SLX4[500-790] were amplified by PCR and cloned into the Ssp1 site of the 438-Sn vector,

in how they are recruited to SMX and how they are regulated in the context of this large assembly.

Combined with prior studies that defined the SLX1-SLX4 and MUS81-EME1-SLX4 interactions in molecular detail[31–34], our study completes the detailed picture of the molecular interactions required for SMX assembly (Fig. 8). Notably, the minimal XPF-SLX4 interaction

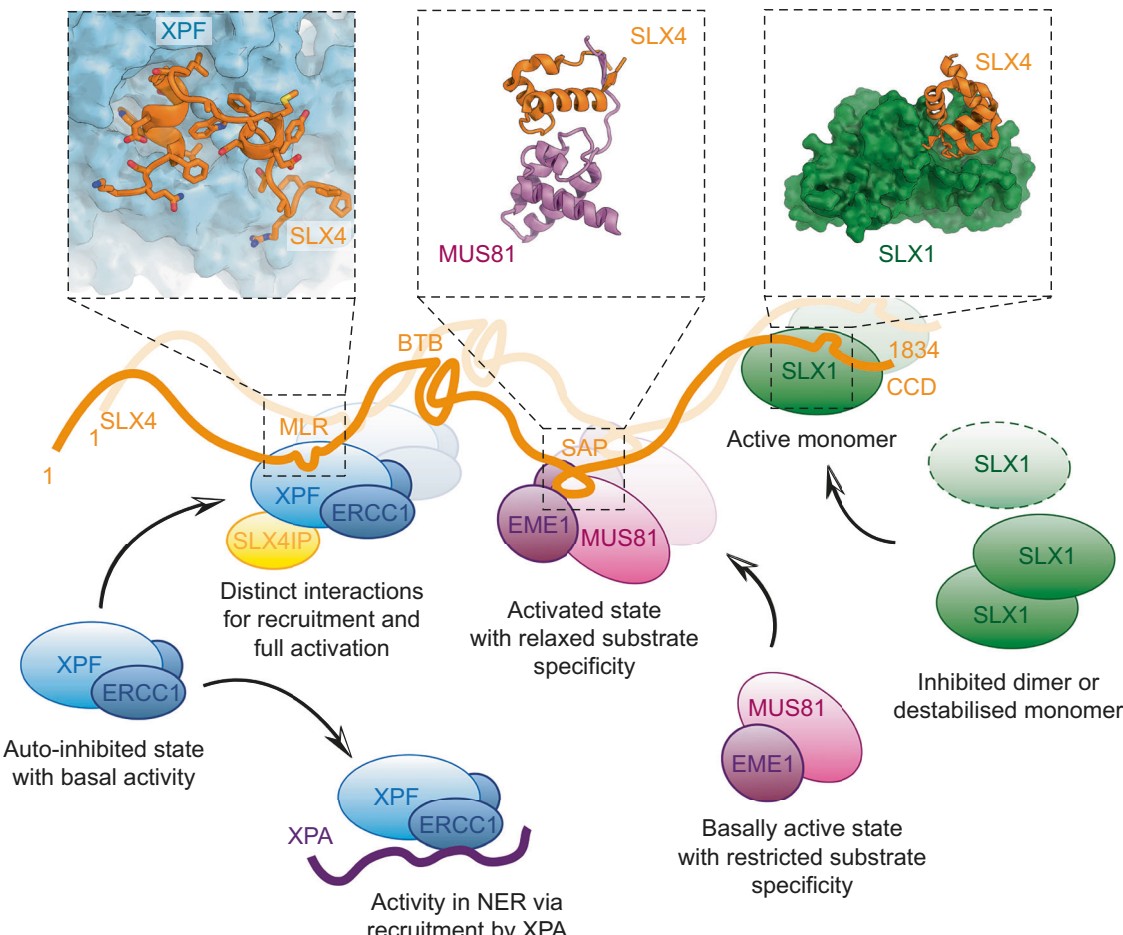

**Fig. 8 | Molecular basis for SMX complex assembly.** Schematic model of the SMX complex shown along with molecular structures of critical contact sites responsible for complex assembly (SLX4-MUS81: PDB ID 7BU5, SLX4-SLX1: PDB ID 4XLG)[34]. A second copy of the complex is shown semi-transparent to represent possible formation of a heterododecameric SMX dimer. Mechanisms for SMX nuclease activation are represented schematically.

which harbours an N-terminal twin-Strep tag, producing 438-Sn-SLX4$^{330-555}$, 438-Sn-SLX4$^{330-555\ 5A}$, 438-Sn-SLX4$^{378-555}$, 438-Sn-SLX4$^{500-555}$, 438-Sn-SLX4$^{330-790}$ and 438-Sn-SLX4$^{500-790}$. The sequences encoding SLX4$^{523-555}$ and SLX4$^{523-555\ 5A}$ were synthesized as gene blocks (gBlock; Integrated DNA Technologies) and cloned into the Ssp1 site of a modified 438-series vector containing a N-terminal MBP tag (referred to as 438-MBP-SLX4$^{523-555}$ and 438-MBP-SLX4$^{523-555\ 5A}$).

## Multi-protein expression constructs
$^{His6}$XPF-ERCC1 and $^{His6}$XPF-ERCC1-SLX4IP expression plasmids (referred to as 438-B-XPF-A-ERCC1 and 438-B-XPF-A-ERCC1-A-SLX4IP, respectively) were generated by step-wise assembly of the individual cassettes expressing one gene each into multi-protein expression constructs. Recipient vectors were linearised using Swa1 restriction enzyme digest, and cassettes for insertion were generated using Pme1 restriction digest. Assembly was performed using In-Fusion Snap Assembly Master Mix (Takara). We did not assemble single plasmids expressing XPF-ERCC1-XPA, XPF-ERCC1-SLX4$^{330-555}$, XPF-ERCC1-SLX4IP-SLX4$^{330-555}$, XPF-ERCC1-SLX4IP-SLX4$^{378-555}$, XPF-ERCC1-SLX4IP-SLX4$^{500-555}$, XPF-ERCC1-SLX4IP-SLX4$^{330-790}$ and XPF-ERCC1-SLX4IP-SLX4$^{500-790}$ as these complexes were generated by co-infection of insect cells with multiple baculoviruses (see below).

## Expression constructs for expression of mutant XPF-ERCC1 complexes
438B-XPF$^{E572R}$-A-ERCC1, 438B-XPF$^{I554P}$-A-ERCC1, 438B-XPF$^{F544A,\ I546A,\ L547A,\ E549K}$-A-ERCC1, 438B-XPF$^{F544A,\ I546A,\ L547A,\ E549K,\ I554P,\ E572R}$-A-ERCC1 were

generated by DNA fragment assembly. 150-bp gBlocks harbouring the designated XPF mutations (Integrated DNA Technologies) were inserted into linearised 438B-XPF-A-ERCC1 plasmid that had been PCR amplified with divergent primers, such as to remove the segment corresponding to the 150-bp gBlock segment, while retaining 20 bp overlap with the termini of the 150-bp mutation-containing fragment. DNA fragments were then assembled using NEBuilder HiFi DNA Assembly (New England Biolabs).

## Protein expression and purification
**Insect cells.** Sf9 (Spodoptera frugiperda) and Hi5 (Trichoplusia ni) cells (both Thermo Fisher Scientific, cat. no. 11496015 and B85502, respectively) were cultured in Insect-XPRESS protein-free medium (Lonza) or Sf-900™ II SFM media (Gibco) at 27 °C.

## Baculoviruses
For generation of recombinant baculoviruses, expression plasmids were transformed into chemically competent E.coli EMbacY cells[57]. Positive bacterial colonies containing the bacmids for protein expression were selected on LB-agar transposition plates containing Gentamicin (7 µg/mL), Kanamycin (50 µg/mL), Tetracycline (10 µg/mL), bluo-gal (100 µg/mL), and IPTG (40 µg/mL). Bacmids were extracted by isopropanol precipitation and transfected into $1 \times 106$ Sf9 cells seeded in a well of a 6-well plate using Cellfectin II (Gibco). At three days post transfection, cells and supernatant were harvested as the first passage (P1) baculovirus, which was then used to infect 30 mL ($5 \times 105$ cells/mL) of Sf9 cells. At least 48 h after cellular proliferation

ceased and when cellular viability dropped below 80%, supernatant was harvested by centrifugation at 2000 × *g* for 10 min and was stored as the P2 virus at 4 °C, supplemented with 2% fetal bovine serum (Gibco). P2 virus was further amplified to generate P3 virus by infecting 30 mL (5 × 105 cells/mL) of Sf9 cells with 1 mL of P2 virus. The P3 virus was harvested when cellular viability dropped below 80% and was used for recombinant protein production in Sf9 cells.

## Protein purification

$^{His6}$XPF-ERCC1-SLX4IP was purified from 1 L of Sf9 cells culture (1 × 106 cells/mL) either infected with 30 mL of P3 $^{His6}$XPF-ERCC1-SLX4IP baculovirus, or co-infected with 25 mL of P3 $^{His6}$XPF-ERCC1 virus and 25 mL of P3 SLX4IP virus. Cells were harvested 72 h post infection when cell viability was around 85%. The cell pellet was resuspended in 40 mL lysis buffer (250 mM KCl, 40 mM HEPES-KOH pH 7.9, 10% glycerol, 2 mM MgCl$_2$, 5 mM β-mercaptoethanol) supplemented with 10 mM Imidazole, home-made protease inhibitors cocktail, and 10 µg/mL DNaseI. Cells were lysed by sonication on ice for 3 min (5 s power on and 10 s power off) using a SONICS Vibra Cell ultrasonic processor at 35% amplitude. Cell lysate was cleared by centrifugation in a Beckman Coulter JA-25.50 fixed-angle rotor at 18,000 rpm for 30 min at 4 °C. The supernatant was mixed with 2.5 mL Ni-NTA Superflow resin (Qiagen) for incubation with gentle rotation at 4 °C for 1 h. The resin was then washed 3 times by lysis buffer supplemented with 25 mM imidazole and eluted in 3 × 5 mL elution buffer (250 mM KCl, 2 mM MgCl$_2$, 40 mM HEPES-KOH pH 7.9, 10% glycerol, 5 mM β-mercaptoethanol, 300 mM imidazole). All three elutions were combined. Buffer was exchanged using a HiPrep Sephadex G-25 26/10 desalting column (Cytiva), which was pre-equilibrated with ion exchange buffer (150 mM KCl, 2 mM MgCl$_2$, 40 mM HEPES-KOH, pH 7.9, 10% glycerol, 5 mM β-mercaptoethanol). Peak fractions were collected, concentrated and loaded on a Capto HiRes Q (Cytiva) anion exchange column pre-equilibrated with ion exchange buffer. XPF-ERCC1-SLX4IP did not bind to the Q column, while XPF-ERCC1, in the absence of SLX4IP, did. Therefore, the trimeric XPF-ERCC1-SLX4IP complex was collected in the flow through, and the dimeric XPF-ERCC1 complex was eluted from the HiRes Q resin with a linear KCl gradient (150 mM-490 mM). Samples were analysed by SDS-PAGE. Fractions containing XPF-ERCC1-SLX4IP or XPF-ERCC1 were pooled separately, concentrated and loaded on a Superdex 200 Increase 10/300GL column (Cytiva), respectively, for size exclusion chromatography. Peak fractions (Supplementary Fig. 2c) were analysed by SDS-PAGE. Selected fractions were pooled, concentrated to 2 mg/mL, flash frozen in liquid nitrogen, and stored at −80 °C.

$^{His6}$XPF$^{E572R}$-ERCC1, $^{His6}$XPF$^{I554P}$-ERCC1, $^{His6}$XPF$^{F544A, I546A, L547A, E549K}$-ERCC1, and $^{His6}$XPF$^{F544A, I546A, L547A, E549K, I554P, E572R}$-ERCC1 were purified by the same four-step purification method and the same buffers used for $^{His6}$XPF-ERCC1-SLX4IP. To test if $^{His6}$XPF$^{mutant}$-ERCC1 could still bind SLX4IP, 1 L Sf9 cells (1 × 106 cells/mL) were co-infected with 25 mL P3 $^{His6}$XPF$^{mutant}$-ERCC1 virus and 25 mL of P3 SLX4IP virus. Cells were harvested, lysed and clarified by centrifugation. The cleared cell lysate was first purified on 2.5 mL Ni-NTA Superflow resin (Qiagen) following the same procedures detailed above. Three elutions were collected and analysed by SDS-PAGE. $^{His6}$XPF$^{mutant}$ lost the ability of binding with SLX4IP, resulting in only $^{His6}$XPF$^{mutant}$-ERCC1 being coarsely purified via the His$_6$ tag. After buffer exchange to ion exchange buffer, peak fractions were concentrated and loaded on a Capto HiRes Q anion exchange column (Cytiva). $^{His6}$XPF$^{mutant}$-ERCC1 bound to the Q column and was eluted with a linear KCl gradient (150 mM-490 mM). Peak fractions were analysed by SDS-PAGE. Selected fractions were pooled, concentrated and loaded on a Superdex 200 Increase 10/300GL column (Cytiva). After analysing the peak fractions by SDS-PAGE, selected fractions were pooled, concentrated to 2 mg/mL, flash frozen in liquid nitrogen, and stored at −80 °C (final samples shown in Supplementary Fig. 9a).

$^{His6}$XPF-ERCC1-SLX4IP-$^{Twin-Strep}$SLX4$^{330-555}$ was purified from 1 L Sf9 cells (1 × 106 cells/mL) co-infected with 25 mL of P3 $^{His6}$XPF-ERCC1-SLX4IP virus and $^{Twin-Strep}$SLX4$^{330-555}$ virus. Cells were lysed as described above. The cleared cell lysate was incubated with 3 mL Strep-Tactin Superflow Plus resin (Qiagen) for 1 h at 4 °C. After washing the resin three times with lysis buffer (250 mM KCl, 2 mM MgCl$_2$, 40 mM HEPES-KOH pH 7.9, 10% glycerol), the protein was eluted in 3 × 3 mL Strep elution buffer (250 mM KCl, 2 mM MgCl$_2$, 40 mM HEPES-KOH pH 7.9, 10% glycerol, 10 mM desthiobiotin). Three elutions were collected, and the buffer was exchanged to ion exchange buffer (150 mM KCl, 2 mM MgCl$_2$, 40 mM HEPES-KOH pH 7.9, 10% glycerol, 5 mM β-mercaptoethanol) using a HiPrep Sephadex G-25 26/10 desalting column (Cytiva). Peak fractions were collected, concentrated and loaded on a Capto HiRes Q anion exchange column (Cytiva) pre-equilibrated with ion exchange buffer. XPF-ERCC1-SLX4IP-SLX4$^{330-555}$ did not bind to the Q column. Therefore, the flow through was collected, while other negatively charged contaminants were removed by the Q column. Subsequently, the flow through was concentrated and loaded onto a Superdex 200 Increase 10/300GL column (Cytiva) for size exclusion chromatography. After analysing the peak fractions by SDS-PAGE, selected fractions were pooled, concentrated to 2 mg/mL, flash frozen in liquid nitrogen, and stored at −80 °C.

$^{His6}$XPF-ERCC1-$^{Twin-Strep}$SLX4$^{330-555}$ (without SLX4IP), $^{His6}$XPF-ERCC1-SLX4IP-$^{Twin-Strep}$SLX4$^{378-555}$, $^{His6}$XPF-ERCC1-SLX4IP-$^{Twin-Strep}$SLX4$^{500-555}$, $^{His6}$XPF-ERCC1-SLX4IP-$^{Twin-Strep}$SLX4$^{330-790}$, and $^{His6}$XPF-ERCC1-SLX4IP-$^{Twin-Strep}$SLX4$^{500-790}$ were purified according to the purification procedure detailed for $^{His6}$XPF-ERCC1-SLX4IP-$^{Twin-Strep}$SLX4$^{330-555}$. For $^{His6}$XPF-ERCC1-$^{Twin-Strep}$SLX4$^{330-555}$, $^{His6}$XPF-ERCC1-SLX4IP-$^{Twin-Strep}$SLX4$^{330-790}$, and $^{His6}$XPF-ERCC1-SLX4IP-$^{Twin-Strep}$SLX4$^{500-790}$, the purification procedure was similar, with the only difference being the Capto HiRes Q anion exchange step. These three complexes bound to the Q column, which was pre-equilibrated with ion exchange buffer (150 mM KCl, 2 mM MgCl$_2$, 40 mM HEPES-KOH pH 7.9, 10% glycerol, 5 mM β-mercaptoethanol). Therefore, they were eluted from the HiRes Q resin with a linear KCl gradient (150 mM−490 mM). Samples were analysed by SDS-PAGE (Supplementary Fig. 2b, e–i). Selected fractions were pooled, concentrated, loaded on a Superdex 200 Increase 10/300GL column (Cytiva), and purified by size exclusion chromatography as described above.

$^{His6}$XPF-ERCC1-$^{His6}$XPA was purified from 500 mL Sf9 cells (1 × 106 cells/mL) co-infected with 25 mL P3 $^{His6}$XPF-ERCC1 virus and P3 $^{His6}$XPA virus. The purification steps were the same as the purification of $^{His6}$XPF$^{mutant}$-ERCC1. A large excess of $^{His6}$XPA from Ni-NTA affinity purification was removed by ion exchange and gel filtration chromatography. $^{His6}$XPA runs at the same position as ERCC1 in a 4−12% NuPAGE Bis-Tris protein gel (Thermo Fisher Scientific), so their bands are not distinguishable in Supplementary Fig. 2a.

## Protein interaction assays by co-expression and co-sedimentation

To compare the interactions of XPF with SLX4$^{330-555}$ or mutant SLX4$^{330-555 \ 5A}$, 30 mL of Sf9 or Hi5 cells (8 ×105 cells/mL) were co-infected with 800 µl of P2 $^{His6}$XPF-ERCC1 virus and 800 µl of P2 $^{Twin-Strep}$SLX4$^{330-555}$ virus. At the same time, the same number of cells were co-infected with 800 µl of P2 $^{His6}$XPF-ERCC1 virus and 800 µl of P2 $^{Twin-Strep}$SLX4$^{330-555 \ 5A}$ virus. At 72 h post infection, cells were harvested and lysed in lysis buffer (250 mM KCl, 40 mM HEPES-KOH pH 7.9, 10% glycerol, 2 mM MgCl$_2$) by freeze-and-thaw using liquid nitrogen. The lysate supernatants were collected by centrifugation at 17,000 g for 15 min, then incubated with 100 µl Strep-Tactin Superflow Plus resin (Qiagen) for 20 min at 4 °C. The Strep-Tactin resin was washed 3 times with lysis buffer by sedimentation of the resin at 300 g in an Eppendorf microtube, removal of buffer, and resuspension of the resin. Bound proteins were eluted with Strep elution buffer (250 mM KCl, 2 mM MgCl$_2$,

40 mM HEPES, pH 7.9, 10% glycerol, 10 mM desthiobiotin). Elutions were analysed by SDS-PAGE.

To compare the interactions of XPF with $^{MBP}$SLX4$^{523-555}$ or mutant $^{MBP}$SLX4$^{523-555\ 5A}$, 30 mL of Sf9 or Hi5 cells ($8 \times 105$ cells/mL) were co-infected with 800 μl of P2 $^{His6}$XPF-ERCC1 virus and 800 μl of P2 $^{MBP}$SLX4$^{523-555}$ virus. At the same time, the same number of cells were co-infected with 800 μl of P2 $^{His6}$XPF-ERCC1 virus and 800 μl of P2 $^{MBP}$SLX4$^{330-555\ 5A}$ virus. At 72 h post infection, cells were harvested and lysed in lysis buffer (250 mM KCl, 40 mM HEPES-KOH pH 7.9, 10% glycerol, 2 mM MgCl$_2$) by freeze-and-thaw using liquid nitrogen. The lysate supernatants were collected by centrifugation at $17,000 \times g$ for 15 min, then incubated with 100 μl amylose resin (New England Biolabs) for 20 min at 4 °C. The amylose resin was washed 3 times by sedimentation of the resin at 300 g in an Eppendorf microtube, removal of buffer, and resuspension of the resin in lysis buffer. Bound proteins were then eluted with MBP elution buffer (250 mM KCl, 2 mM MgCl$_2$, 40 mM HEPES, pH 7.9, 10% glycerol, 10 mM maltose). Elutions were analysed by SDS-PAGE.

### Endonuclease assays

A Y-shaped DNA substrate containing a Cyanine3 (Cy3) fluorescent label at the 3′-terminus of one of the single-stranded segments (designated as the top strand; Fig. 3a) was used for endonuclease assays. To prepare the Y-shaped DNA substrate, the top and bottom stands were dissolved in Annealing Buffer (10 mM Tris-HCl, pH 7.5, 50 mM NaCl, 1 mM EDTA) at a concentration of 10 μM, then denatured at 98 °C for 5 min and annealed by gradually lowering the temperature to 4 °C over 3 h (approximately 0.5 °C/min) at a molar ratio of 1:1.5.

For the enzymatic assay, purified protein complex and annealed DNA substrate were diluted in Nuclease Activity Buffer (5 mM HEPES pH 7.9, 50 mM KCl, 10% Glycerol, 0.5 mM DTT, 0.15 mg/mL BSA, 1.5 mM MnCl$_2$) to 150 nM (protein) and 300 nM, 600 nM, or 900 nM (DNA), respectively, which equals to enzyme:DNA molar ratios of 1:2 (Supplementary Fig. 9b), 1:4 (Fig. 3 and Supplementary Fig. 5), or 1:6 (Fig. 5, Supplementary Fig. 7). Reaction mixtures were incubated at 30 °C. After 0 min, 5 min, 10 min, and 30 min of incubation, reactions were quenched by adding Novex TBE-Urea Sample Buffer (2x) (Thermo Fisher Scientific) and boiled at 100 °C for 3 min. All samples were loaded onto either 15% 15-well Novex TBE-Urea PAGE gels (Thermo Fisher Scientific) or 15% 26-well Criterion TBE-Urea PAGE gels (Bio-Rad). DNA fragments were separated by electrophoresis at a constant voltage of 300 V for 30 min in 1x TBE buffer and visualized using a LICOR ODYSSEY M imaging system.

### Cryo-EM sample preparation

UltrAuFoil R1.2/1.3 300 mesh gold grids (Quantifoil Microtools GmbH) were plasma cleaned for 50 s using a Tergeo EM Plasma Cleaner (PIE Scientific). Purified XPF-ERCC1-XPA, XPF-ERCC1-SLX4IP and XPF-ERCCC1-SLX4IP-SLX4$^{330-555}$ were diluted five- to ten-fold from approximately 2 mg/mL stock into cryo-EM buffer (20 mM HEPES-KOH pH 7.9, 2 mM MgCl$_2$, 200 mM KCl, 5 mM β-mercaptoethanol) before addition onto the grids. 4 μL of sample was applied to the grid, blotted for 1, 1.5, 1.5 or 2 s (eight grids for each complex) with blot force 0 at 4 °C and 100% humidity using a Vitrobot Mark IV (Thermo Fisher Scientific), and plunge-frozen in liquid ethane cooled by liquid nitrogen.

For grid preparation of XPF-ERCC1-SLX4IP-SLX4$^{330-555}$ with Y-fork DNA substrate, the two DNA strands with phosphorothioate modifications near the preferred XPF cleavage site (Fig. 3d) were dissolved and annealed as described above, at a molar ratio of 1:1. Reaction mixtures containing 2.5 μM annealed fork DNA and 1.25 μM purified XPF-ERCC1-SLX4IP-SLX4$^{330-555}$ in Cryo-EM DNA Substrate Buffer (20 mM HEPES-KOH pH 7.9, 200 mM KCl, 2 mM MnCl$_2$) were incubated on ice for 2 min, then applied to cryo-EM grids and blotted using the same Vitrobot settings described above.

All grids were later clipped into autogrid cartridges (Thermo Fisher Scientific) for use in autoloader cryo-EM systems.

### Cryo-EM data collection

**XPF-ERCC1-XPA.** For structure determination of XPF-ERCC1-XPA, grids were loaded into a 300 kV Titan Krios G3i cryo-transmission electron microscope (cryo-TEM; Thermo Fisher Scientific) equipped with a K3 direct electron detector and a BioQuantum energy filter (Gatan; operated with Digital Micrograph version 3.53.41360). Grid atlases were acquired using EPU (Thermo Fisher Scientific, version 3.4), and one grid was selected for data acquisition. Grid squares were manually selected and automatically brought to eucentric height. Holes were automatically detected and selected using ice thickness filters in EPU. For large-scale data collection, the microscope was set to 165,000-fold magnification and electron micrograph movies were collected in TIFF format with a nominal pixel size of 0.52 Å/pixel, a total electron exposure of 60 e$^-$/Å$^2$, and a defocus range of −0.8 μm to −2.5 μm. Aberration-free image shift (AFIS) and fringe-free imaging (FFI) were employed to accelerate data acquisition to approximately 650 movies/hour. In total, 12,629 TIFF format movies (50 frames/movie) were acquired on a single grid.

### XPF-ERCC1-SLX4IP

Data acquisition for XPF-ERCC1-SLX4IP was performed using a Glacios cryo-TEM (Thermo Fisher Scientific) operating at 200 kV acceleration voltage and equipped with a Falcon 4i direct electron detector (Thermo Fisher Scientific). After atlas acquisition, the quality (ice thickness, square integrity) of the grids was assessed, and one grid was selected for a short collection of 293 EER format movies, which were processed on-the-fly using cryoSPARC live[40] to assess data quality and particle orientations by 2D classification. After assessing the 2D class averages, data acquisition was continued on this grid. For data collection, the microscope was set to 240,000-fold nominal magnification. Due to restricted camera offload server throughput, the format of the collected electron micrograph movies was changed from EER to MRC format and the frame number later reduced from 50 to 34 frames for large-scale data collection. Movies were collected with a pixel size of 0.5675 Å/pixel (246,669-fold calibrated magnification), a total electron exposure of 60 e$^-$/Å$^2$ fractionated into 50 or 34 movie frames, and a defocus range of −0.7 μm to −1.7 μm. AFIS was employed to accelerate data acquisition to approximately 280 movies/hour. In total, 6,894 MRC format movies were acquired.

### XPF-ERCC1-SLX4IP-SLX4$^{330-555}$

For structure determination of XPF-ERCC1-SLX4IP-SLX4$^{330-555}$, grids were initially screened using a Glacios cryo-TEM equipped with a Falcon 4i direct electron detector (Thermo Fisher Scientific) and grid quality was assessed. For high-resolution structure determination, the selected grid was loaded into a 300 kV Titan Krios G3i cryo-TEM (Thermo Fisher Scientific) equipped with a K3 direct electron detector and BioQuantum energy filter (Gatan; operated with Digital Micrograph version 3.53.41360). The microscope was set to 165,000-fold magnification, and electron micrograph movies were collected in TIFF format with a nominal pixel size of 0.52 Å/pix, a total electron exposure of 60 e$^-$/Å$^2$, and a defocus range of −1.0 μm to −2.5 μm, using EPU (Thermo Fisher Scientific, version 3.4). AFIS and FFI were employed to accelerate data acquisition to approximately 650 movies/hour. In total, 14,573 TIFF format movies (50 frames/movie) were acquired.

### DNA-bound XPF-ERCC1-SLX4IP-SLX4$^{330-555}$

For Y-fork DNA-bound XPF-ERCC1-SLX4IP- SLX4$^{330-555}$ data acquisition, two pre-screened grids were loaded into a 200 kV Glacios cryo-TEM equipped with a Falcon 4i direct electron detector and a Selectris imaging filter (Thermo Fisher Scientific). In addition to the data acquisition settings used for collection of the XPF-ERCC1-SLX4IP data

on the Glacios, the Selectris energy filter was used with a 10 eV slit width, and the microscope was set to 165,000-fold nominal magnification. Electron micrograph movies were collected in EER format with a pixel size of 0.7 Å/pixel, a total electron exposure of 60 e⁻/Å², and a defocus range of −0.4 µm to −1.7 µm. FFI, AFIS, and EPU Multigrid (Thermo Fisher Scientific, version 3.8.1) were used to accelerate data collection on two grids to approximately 500 movies/hour. In total, 11,694 EER format movies were acquired (1641 movies on grid I, 10,053 movies on grid II).

### Image processing

**General data processing strategy and data processing for XPF-ERCC1-XPA**. Image processing for reconstruction of the XPF-ERCC1-XPA complex is detailed in Supplementary Fig. 10. The dataset comprising 12,629 electron micrograph movies in TIFF format was initially pre-processed in cryoSPARC live and cryoSPARC v3.3.1[40] and then refined in RELION 4.0 beta[41]. In cryoSPARC live, movies were motion corrected and subsequently triaged based on quality of the CTF fit, relative ice thickness, beam-induced motion, and number of particles picked, resulting in the retention of 9174 high-quality movies. Particles were selected using blob picking and extracted in $360 \times 360$ pixel boxes, binned to $180 \times 180$ pixels (resulting in a nominal pixel size of 1.04 Å, later re-calibrated to 1.02 Å per pixel). Based on streaming 2D classification (50 classes) and using a previous reconstruction of XPF-ERCC1 (EMD-10337)[24] as an initial reference, 746,639 particles were selected and refined to 3.4 Å resolution, as assessed by Fourier Shell Correlation (FSC) at the 0.143 cut-off[60], within cryoSPARC live.

To maximise the retrieval of high-quality particles, additional particle selection and classification strategies were employed within cryoSPARC after the end of the cryoSPARC live session, similar to previously reported workflows[61]. First, the 746,639 particles images selected during live processing were re-classified (200 classes, batch size of 200 particles per class, 100 iterations), and 602,418 particle images were retained (particle set i). The result of this re-classification also provided templates for template picking and high-quality particles for Topaz picking (see below). Second, all 3,692,251 particles blob-picked in cryoSPARC live were re-classified (200 classes, batch size of 200 particles per class, 100 iterations). From this classification, 662,339 particle images were selected (particle set ii). Third, a set of eight high-quality templates from 2D classification was used as references for template picking. 4,185,747 template-picked particle images were subjected to 2D classification (200 classes, batch size of 200 particles per class, 100 iterations), and 533,936 particle images were retained (particle set iii). Fourth, a set of approx. 100,000 high-quality particle images were used to train a Topaz model[62], and 2,041,880 Topaz-picked particles were 2D classified (200 classes, batch size of 200 particles per class, 100 iterations). Of these 727,733 particles images were retained (particle set iv).

The particles from all picking strategies (particle sets i–iv) were combined, and duplicated particles (identified by more than one picking strategy) were removed. After re-extraction of the resulting 1,280,280 particle images with re-centering, a final 2D classification (100 classes, batch size of 200 particles per class, 100 iterations) was performed, and 923,599 particles were selected. Duplicates within these particles were again removed, the particles were re-extracted with re-centering, duplicates were removed again, and 920,767 particle images were retained. The particle coordinates were exported and converted to a RELION-format *.star file using PyEM programs[63], followed by adjustments to adhere to RELION conventions. This general cryoSPARC pre-processing pipeline was applied to all datasets collected in this work.

After import of the particle location information into RELION 4.0 (ref. [41]), 920,767 particle images were extracted at $180 \times 180$ pixel box size from two-fold binned micrographs that had been motion corrected using the implementation of a MOTIONCOR2-like algorithm in RELION[64]. An initial 3D auto-refined yielded a 3.5 Å-resolution map of the XPF-ERCC1-XPA complex. After alignment-free 3D classification using 4 classes (regularization parameter $\tau = 20$, 35 iterations), a class of 73,737 particle images was selected and 3D auto-refined to 3.3 Å resolution. Bayesian polishing (trained on the initial 3D auto-refinement in order to ensure the presence of a sufficient number of particles per micrograph)[65] and 3D auto-refinement resulted in a 3.0 Å-resolution map. CTF refinement[66] to correct for beam tilt, three-fold astigmatism, and 4th-order aberrations yielded the final 2.9 Å-resolution map of the XPF-ERCC1-XPA complex, which was post-processed using the re-calibrated pixel size of 1.02 Å and used for interpretation.

### XPF-ERCC1-SLX4IP-SLX4[330-555]

Cryo-EM data of XPF-ERCC1-SLX4IP-SLX4[330-555] were processed in cryoSPARC live and cryoSPARC V4.4.1[40] and RELION 4.0 beta[41] according to the general strategy outlined above (Supplementary Fig. 11). Particles selected from 9,970 accepted micrographs were refined to 3.5 Å using streaming 3D reconstruction in cryoSPARC live. After 2D-classification and curation of cryoSPARC live (blob picked), template picked, and Topaz picked particles, 1,134,962 particles were retained and subsequently imported into RELION 4.0 beta for downstream processing.

In RELION 4.0 beta, 14,573 TIFF format movies were imported and motion corrected with a binning factor of 2 (50 movie frames, 1.2 e⁻/Å² per frame). 1,134,962 particles were re-extracted from the motion-corrected micrographs with $180 \times 180$ pixel box size. Particles were then subjected to masked 3D auto-refinement and a 3D classification with no alignment (25 iterations, $\tau = 20$). Particles from two bad classes were removed. The remaining particles were 3D auto-refined, subjected to Bayesian polishing[65], and 3D auto-refined again. At this stage, only a small and fragmented segment of SLX4[330-555], corresponding to approximately residues 527–552, was visible on the cryo-EM map. Therefore, focused classification with background signal subtraction[67] was performed, aiming to better resolve the visualised SLX4 segment and its interface with XPF. Alignment-free, signal subtracted 3D classification (35 iterations, $\tau = 32$, no alignment) yielded 261,803 particles from one class that exhibited more complete density for SLX4 residues 527–552. To improve map quality after auto-refinement, this particle subset was subjected to an additional round of 3D classification (20 iterations, $\tau = 24$, no alignment). 22,617 particles were selected, auto-refined, and post-processed to 3.2 Å.

### DNA-bound XPF-ERCC1-SLX4IP-SLX4[330-555]

Cryo-EM data for Y-fork DNA bound XPF-ERCC1-SLX4IP-SLX4[330-555] were initially pre-processed in cryoSPARC live and cryoSPARC v4.4.1[40], and then transferred to RELION 5.0 beta for further processing (Supplementary Fig. 12). The data collected from two grids were initially processed separately, using the general pre-processing strategies described above. Compared to the other datasets, only three particle sets were derived from each dataset. High-quality blob-picked particles were obtained by re-classification of all blob-picked particles from the cryoSPARC live session (particle sets i/iv for the two grids, respectively). Subsequently, high-quality particles from the re-classification were used as templates for template picking (particle sets ii/v) and training resource for Topaz picking (particle sets iii/vi). Particles retrieved by all three picking algorithms (particle sets i-iii for the first grid, particle sets iv-vi for the second grid) were combined, re-extracted and re-centered. After the removal of duplicate particles, 763,862 particles were retained, which were subsequently exported for use in RELION 5.0 beta.

In addition to particle curation, cryoSPARC live was also used for generation of initial models for RELION 3D classification and 3D auto-refinement (Supplementary Fig. 12). More specifically, a small number of particles (fewer than 50,000) from good 2D classes were

homogeneously refined using our XPF-ERCC1-SLX4IP-SLX4[330-555] structure. Subsequently, the 3D volume obtained from this homogenous refinement was used as a reference for heterogeneous refinement (two references) of the same small particle subset. The resulting 3D volumes were subsequently used as references for heterogeneous refinement (two references) of 515,712 high-quality particles (490,739 from grid II, 24,976 from grid I), which resulted in separation of DNA-bound and apo classes. The 246,363 particles assigned to the DNA-bound class were homogenously refined to 3.7 Å. This 3D volume was used as the initial model in RELION.

In RELION 5.0 beta, 11,694 EER format movies successfully processed in cryoSPARC live were imported and motion corrected with a binning factor of 2. 21 raw EER frames were grouped into one fraction for motion correction, resulting in 59 fractions and a dose of 1.02 e⁻/Å² per fraction. 763,862 particles were re-extracted from the motion-corrected micrographs with 140 ×140 pixels box size. These particles were first subjected to 3D classification with alignment (2 classes, 25 iterations, $\tau = 4$, with BLUSH regularisation[68]) using the initial reference from cryoSPARC. This classification recapitulated the split into DNA-bound and apo-classes observed in cryoSPARC. The class containing 473,334 particles without bound DNA was 3D auto-refined (BLUSH regularisation[68] was used for this and all subsequent refinements) and 3D-classified (2 classes, $\tau = 4$, with BLUSH). 72,254 selected particles were 3D auto-refined to yield a reconstruction of the DNA-free complex (3.8 Å resolution).

The DNA-bound class containing 290,528 particles was 3D auto-refined, CTF refined[66] with 4th order aberrations enabled, and 3D auto-refined. Those particles underwent two rounds of Bayesian polishing[65] with 3D auto-refinement in between polishing steps and with an enlargement of the extraction box to 320 × 320 pixels in the un-binned movies, which were down-scaled to 196 ×196 pixels, resulting in a pixel size of 1.14 Å. 3D auto-refinement and alignment-free 3D classification (3 classes, $\tau = 20$, no BLUSH) were performed, and 55,779 particles were selected, 3D auto-refined, and post-processed to yield the final 3.4 Å-resolution map.

## XPF-ERCC1-SLX4IP

The XPF-ERCC1-SLX4IP data were initially pre-processed in cryoSPARC live and cryoSPARC V3.3.1[40] and then migrated to RELION 4.0 beta[41] for further processing and 3D reconstruction (Supplementary Fig. 13). Processing followed the same general strategy outlined above, with minor modifications. In cryoSPARC live only the first 32 frames of the acquired MRC-format raw movies were considered for motion correction with 2x binning. Subsequent micrograph curation resulted in 6003 micrographs accepted and 891 rejected micrographs. Following steaming 2D classification, 328,333 particles from five good 2D classes were selected and refined to 3.6 Å resolution in streaming homogenous refinement using a previous reconstruction of XPF-ERCC1 (EMD-10337)[24] as an initial reference.

Further processing in cryoSPARC was performed as described in the general procedure above. Reclassification of selected blob picking particles and all blob picking particles yielded 251,381 (particle set i) and 358,603 (particle set ii) particles. Template picking and Topaz picking followed by 2D classification yielded 372,312 (particle set iii) and 356,468 particles (particle set iv). Particle sets i-iv were joined, duplicates were removed, the combined particles were re-extracted and re-centred, and duplicates generated during these processes were removed again. In total, 613,225 particles from all picking strategies were exported and particle coordinates were converted to a RELION-compatible *.star file as described above.

In RELION 4.0 beta, 6894 MRC format movies successfully processed in cryoSPARC live were imported and motion corrected with a binning factor of 2 by the RELION implementation of a MOTIONCOR2-like algorithm[64]. Because of the presence of a mixture of 34-frame and 50-frame movies, dose weighting at this stage was inaccurate for some of the movies, but this was corrected at the particle polishing stage (see below). Particles were re-extracted from the motion-corrected micrographs with 180 × 180 pixels box size using coordinate information contained in the imported particles from cryoSPARC. In total, 613,225 particles were extracted. Particles were then subjected to masked 3D auto-refinement using the cryoSPARC live reconstruction as an initial model and 3D classification with no alignment (15 iterations, $\tau = 20$). 107,603 particles from the best 3D class were selected and 3D auto-refined to 3.6 Å resolution. To further improve the map quality, those particles underwent two rounds of CTF refinement[66] and 3D auto-refinement, followed by Bayesian polishing[65]. For Bayesian polishing, particle subsets extracted from those movies collected with 34 and 50 frames were treated separately, and polished particles were re-joined after polishing. After an additional round of 3D auto-refinement and CTF refinement, the polished particles were 3D auto-refined twice (the second time with an optimised mask and matching reference volume) and post-processed, providing the final to 3.2 Å-resolution reconstruction.

## Other computational procedures

**AlphaFold2 structure predictions.** An installation of AlphaFold2 was run in multimer mode[38,39] on a GPU workstation. Due to memory limitations, predictions containing SLX4 were run with the N- and C-terminal halves of the protein (i.e. residues 1–800 and 801–1834) separately. Subsequent predictions (run after the completion of cryo-EM structure determination of the XPF-ERCC1-SLX4IP-SLX4[330-555] complex) with AlphaFold3[69], incorporating the entire complex in a single run did not show meaningful differences to the earlier AlphaFold2 predictions on which analysis and construct design were based.

## Atomic model building and refinement

Atomic models were built by iterative re-building in COOT[70] and coordinate refinement in PHENIX[71], based in initial models obtained from AlphaFold2 Multimer[39] or AlphaFold3[69] and the previous cryo-EM structure of XPF-ERCC1 (PDB ID 6SXA)[24]. Final refinement of the atomic coordinates was performed in PHENIX, and the resulting structures were validated using MOLPROBITY as implemented in PHENIX. Refinement statistics are provided in Supplementary Tables 1 and 2.

## Molecular graphics

Depictions showing molecular models and cryo-EM maps were created using UCSF ChimeraX[72] and PyMOL (The PyMOL Molecular Graphics System, version 3.7, Schroedinger LLC).

## Experiments in human cells

**Human cell lines and compound treatments.** HEK293TN (Systems Biosciences, cat. no. LV900A-1; RRID:CVCL_UL49) cell lines were cultured in Dulbecco's modified Eagle's medium (DMEM) supplemented with 10% fetal bovine serum (FBS) and standard antibiotics. RPE p53-/-FRT/TR cells[73,74] were obtained as a gift from S. Jackson. Cell lines were regularly tested to confirm the absence of mycoplasma contamination.

**Plasmids for assays in human cells.** The human XPF coding sequence was transferred from a 438-series insect cell expression vector to pcDNA5-Neo-FRT/TR (RRID:Addgene_41000; a gift from Dr. J. Mansfeld) via HiFi DNA assembly. pcDNA5-Neo-FRT/TO-eGFP-XPF expression constructs containing the I554P and E572R mutations were generated by subjecting the pcDNA5-FRT/TO-Neo-eGFP-XPF plasmid to site directed mutagenesis using the Q5 site directed mutagenesis kit (New England Biolabs cat. No. E0554S). The pcDNA5-Neo-FRT/TO-eGFP-XPF construct containing the M7 mutation (F554A, I546A, L545A and E549K) or the X6 mutation (F554A, I546A, L545A, E549K, I554P and E572R) were generated via HiFi assembly of the pcDNA5-Neo-FRT/TO-eGFP-XPF vector to replace the surrounding cDNA sequence with a

gBlock fragment (Integrated DNA Technologies) containing the mutated DNA sequence. For amplification, plasmids with the pCDNA5 backbone were transformed and amplified in DH5α bacteria (Thermo Scientific, cat. No. EC0112).

**GFP-trap agarose co-immunoprecipitation.** HEK293TN cells were seeded to be at approximately 25-30% confluence the next day in a 15 cm tissue culture dish. The following day cells were transfected using Lipofectamine 2000 transfection reagent (Thermo Scientific) with 24 μg of plasmid DNA encoding eGFP-XPF cassettes following manufacturer's instructions.

48 hrs post transfection cells were harvested, snap frozen on dry ice and stored at -80°C. Cell pellets were removed from -80°C storage and thawed on ice in 2.4 mL IP buffer 1 (100 mM NaCl, 0.2% Igepal CA-630, 1 mM MgCl2, 10% glycerol, 5 mM NaF, 50 mM Tris-HCl, pH 7.5), supplemented with EDTA Free SIGMAFAST protease inhibitor (Sigma Aldrich) and 25 U/mL Benzonase (Novagen). Cells were resuspended and rotated at 4°C for 90 min. Benzonase was then inhibited by adjustment of the NaCl concentration to 200 mM and 2 mM EDTA and rotated for a further 30 min at 4°C on a carousel. The cell suspension was then centrifuged at 16,000 g for 25 min at 4°C. Then, 20 μl binding control bead slurry was washed with 500 μl IP buffer 2 (200 mM NaCl, 0.2% Igepal CA-630, 1 mM MgCl2, 10% glycerol, 5 mM NaF, 2 mM EDTA, 50 mM Tris-HCl, pH 7.5). The binding control beads were then centrifuged at 2,000 g for 2 min. The supernatant was then removed and the IP2 wash was repeated twice more. Cell lysates were then added to the agarose binding control beads in 15 mL centrifuge tubes and subsequently rotated at 4°C on carousel for 60 min. The agarose binding control beads were then pelleted by centrifugation at 2000 g for 2 minutes, the supernatant was then transferred to a new tube. BCA assay was then carried out to determine the protein concentration of the pre-cleared cell lysate. Subsequently, 20 μl GFP_TRAP_A bead slurry was washed with 500 μl IP buffer 2 (200 mM NaCl, 0.2% Igepal CA-630, 1 mM MgCl2, 10% glycerol, 5 mM NaF, 2 mM EDTA, 50 mM Tris-HCl, pH 7.5). The beads were then centrifuged at 2,000 g for 2 min. The supernatant was removed and the IP2 wash step is repeated twice more. Cell lysates were then diluted to 1 mg/mL final concentration in IP buffer 2 and added to the washed GFP_TRAP_A beads in 15 mL conical tubes and rotated at 4 °C on carousel for 120 min. The 15 mL conical tubes were then centrifuged at $2000 \times g$ for 2 min at 4 °C, and the supernatant was removed. The beads were then washed with 500 μl IP2 buffer and transferred to a 1.5 mL microcentrifuge tube, then centrifuged at $2000 \times g$ for 2 min at 4 °C. The supernatant was removed, and the IP2 wash was repeated twice more. Bound protein complexes were eluted from the GFP_TRAP_A beads by addition of 50 μl 2x SDS buffer (120 mM Tris/Cl, pH 6.8, 20% glycerol, 4% SDS, 0.04% bromophenol blue, 10% β-mercaptoethanol) followed by incubation at 95 °C for 10 min. Tubes were then placed on ice for 10 min followed by centrifugation at $2000 \times g$ for 2 min, and the supernatant was transferred to a new 1.5 mL microcentrifuge tube ready for downstream western blot analysis.

**Immunoblotting**

Co-IP samples were resolved by SDS-PAGE and transferred to nitrocellulose membrane followed by blocking in 5% low fat milk in 1x TBS/0.1% Tween-20 for 1 h at room temperature. Membranes were washed 3 times for 5 min in 1x TBS/0.1% Tween-20 and incubated overnight at 4 °C in the indicated primary antibodies in 5% low fat milk in 1x TBS/0.1% Tween-20. Membranes were subsequently washed 3 times for 5 min in 1x TBS/0.1% Tween-20 and incubated in 5% low fat milk in 1x TBS/0.1% Tween-20 containing secondary antibodies for 1 h at room temperature. Membranes were again washed 3 times for 5 min in 1x TBS/0.1% Tween-20 and developed using Immobilon Western HRP Substrate (Millipore, WBKLS0S00) and imaged using the Azure C280, 300 or 600 instruments (Azure Biosystems).

Primary antibodies used were: GFP (Roche Cat# 11814460001, RRID:AB_390913, 1:500), MUS81 (Santa Cruz Biotechnology Cat# sc-47692, RRID:AB_2147129,1:500), ERCC1 (Santa Cruz Biotechnology Cat# sc-17809, RRID:AB_2278023,1:500), SLX4 (MRC-PPU Cat# S714C, RRID:AB_2752254, 1:500), SLX1 (Proteintech Cat# 21158-1-AP, RRID:AB_2752255, 1:500), EME1 (Santa Cruz Biotechnology Cat# sc-393363, 1:500), XPF (Bethyl Cat# A301-315A, RRID:AB_938089 1:500), and SLX4IP (Santa Cruz Biotechnology Cat# sc-377066, RRID:AB_2752253).

Secondary antibodies used were anti-mouse IgG-HRP (Dako, P0447, 1:2000), anti-rabbit IgG-HRP (Dako, P0448, 1:5000) and anti-sheep IgG-HRP (Abcam Cat# ab6747, RRID:AB_955453, 1:1000).

Uncropped membranes for Western blots shown in Figs. 2d and 7b, and Supplementary Fig. 9c are shown in Supplementary Fig. 14–16.

**Generation of XPF (ERCC4) knock-out cells by CRISPR ribonucleoprotein delivery**

RPE p53-/- FRT/TR cells[73,74] (a gift from S. Jackson) were subjected to CRISPR/Cas9-mediated mutagenesis via Cas9/sgRNA ribonucleoprotein delivery. Alt-R Streptococcus pyogenes (S.p.) Cas9-GFP V3 (Integrated DNA Technologies, cat. no. 10008100), Alt-R CRISPR-Cas9 tracrRNA, ATTO™ 550 (Integrated DNA Technologies, cat. no. 1075927) and Alt-R CRISPR-Cas9 crRNA targeting exon 1 of the CIP2A locus (ERCC4 crRNA: TGTCGCTCGTACTCCAGCAG) were procured from Integrated DNA Technologies. Subsequently, crRNA and tracrRNA were resuspended in nuclease free duplex buffer at 100 μM concentration. Then 1 μl of each of 100 μM crRNA and tracRNA were diluted in 98 μl of nuclease free duplex buffer, mixed and denatured at 95 °C for 5 min followed by gradient cooling to room temperature, to facilitate crRNA and tracrRNA annealing to form sgRNA. S.p. Cas9 Nuclease V3 was vortexed vigorously and diluted to 1 μM in Optimem (Gibco, cat. no. 31985062). At which point 24 μl of 1 μM sgRNA was combined with 24 μl of 1 μM Cas9 and 20 μl of Lipofectamine RNAimax (Invitrogen, cat. no. 13778075) in 732 μl of Optimem to generate an 800 μl RNP transfection mixture. The RNP transfection mixture was incubated at room temperature for 20 min to facilitate complex formation. In this time, RPE1 p53-/- FRT/TR cells were trypsinised and diluted to $4 \times 10^5$ cells/mL. Subsequently, 800 μl of the RNP transfection mixture was added to a well of a 6-well tissue culture dish, followed by addition of 1600 μl of the diluted cell suspension. The cells and RNP mixture were briefly mixed by pipetting and left to incubate at 37 °C with 5% $CO_2$ for 24 hours. The cells were then trypsinised and eGFP and ATTO550 double positive cells were single-cell fluorescence activated cell sorted using the BD FACSymphony S6 Cell Sorter into 96 well tissue culture plates and incubate at 37 °C with 5% $CO_2$ for 14 days. At which stage colonies were transferred to 6-well plates and incubated at 37 °C with 5% $CO_2$ for further 5 days. Then, colonies were trypsinised, and 90% of the cell suspension was isolated by centrifugation and processed by Western blotting to detect ERCC4 loss at the protein level (Supplementary Fig. 9d). Uncropped membranes for Western blots shown in Supplementary Fig. 9d are shown in Supplementary Fig. 17.

Successful knock-out was additionally verified by amplification of the region around the first exon of the ERCC4 gene by PCR (primers: 5'-CTG CGA CCC GGA AGA GCT TCC ATG GAG TCA GGG CAG CCG GCT-3', 5'-CGC AGT GTG AGG GAC CTG CAT CCC CCT GGG GAC CCC TGC CAT CCT TCT CTG TGT-3') and deep sequencing (using the GENEWIZ commercial service provided by Azenta Life Sciences). Sequencing data were analysed using the CRISPResso2 web server[75].

**Complementation of XPF knock-out cells. Inducible cell line generation**

To facilitate Flp-In recombination of pCDNA5-FRT/TO-Neo-eGFP, pCDNA5-FRT/TO-Neo-eGFP-XPF^WT, or pCDNA5-FRT/TO-Neo-eGFP-

XPF$^{544\text{-}549\ 3AK}$, 2 μg of the Flp-Recombinase expressing plasmid pOG44 was co-transfected with 6 μg of the respective pcDNA5-FRT/TO plasmid into RPE1 p53$^{-/-}$ FRT/TR cells, using Lipofectamine 3000 (Invitrogen, cat. no. L3000001) following the manufacturer's guidelines. Cells were incubated for 24 h at 37 °C with 5% $CO_2$ followed by addition of 1 mg/mL G418 (Invivogen, cat. no. ant-gn-1) to facilitate antibiotic selection. G418 was replaced every 72 h. After 14 days, 100 ng/mL of doxycycline was added to the medium, and the cells were incubated a further 24 h at 37 °C with 5% $CO_2$. eGFP expressing cells were sorted by fluorescence activated cell sorting (FACS) using the BD Symphony S6 Cell Sorter, gating for eGFP positive cells. Cells were maintained in G418-containing medium (500 μg/mL) during continuous culture.

### Cell Titer-Glo survival assay

Cell Titer-Glo (CTG) luminescent cell viability assays (Promega) were performed in accordance with the manufacturer's recommendations. 250 cells per well in 96-well plates were incubated in 100 ng/mL doxycycline media alone or with indicated doses of cis-platin (Sigma Aldrich) for 5 days at 37 °C with 5% $CO_2$. A working solution of CTG reagent was prepared by dilution of reagent 1:4 in 1x PBS. Subsequently, cell culture medium was aspirated and 50 μl 1x CTG working solution was added to wells. Subsequently, plates were incubated in the dark on a plate rocker for 10 min at RT. Luminescence was measured using a Victor X5 plate reader (Perkin Elmer) running the Perkin Elmer 2030 software (version 4.0). The background fluorescence intensity measured from wells containing media only was subtracted from measurements from wells containing cells. Technical replicates were measured in different wells; biological replicates were independent experiments. Fluorescence intensity was normalised to measurements from wells containing cells only, without the addition of cisplatin, and the relative percentage of intensity measurements was plotted in PRISM (version 10.6.1; GraphPad Software).

### Statistics and reproducibility

Conclusions from small-scale co-sedimentation and co-immunoprecipitation assays were validated by one additional biological replicate (Figs. 2b, c, d, and 7b). The cis-platin survival assay (Fig. 7c) is based on three biological replicates (see above). Experiments for analysis of XPF-ERCC1 activation by endonuclease assays (Figs. 3b, c and 5c) are comprised of three technical replicates, independently performed as separate experiments. Uncropped Western blot membranes along with uncropped SDS-PAGE and Urea-PAGE gels are provided as Source Data.

### Reporting summary

Further information on research design is available in the Nature Portfolio Reporting Summary linked to this article.

## Data availability

Requests for materials should be addressed to the corresponding author. The cryo-EM maps and atomic coordinate models for the XPF-ERCC1-XPA, XPF-ERCC1-SLX4IP-SLX4$^{330\text{-}555}$, and XPF-ERCC1-SLX4IP-SLX4$^{330\text{-}555}$-DNA complexes were deposited to the Electron Microscopy Data Bank (EMDB) and Protein Data Bank (PDB) with accession codes EMD-53054, EMD-53055, and EMD-53058, and PDB-9QEC [https://doi.org/10.2210/pdb9QEC/pdb], PDB-9QED [https://doi.org/10.2210/pdb9QED/pdb], and PDB-9QEE [https://doi.org/10.2210/pdb9QEE/pdb], respectively. The cryo-EM maps of the XPF-ERCC1-SLX4IP complex and the DNA-free XPF-ERCC1-SLX4IP-SLX4$^{330\text{-}555}$ complex obtained from the same grid as the DNA-bound complex were deposited to the EMDB with accession codes EMD-53061 and EMD-53059. The AlphaFold Multimer model of the XPF-ERCC1-SLX4IP-SLX4 complex has been deposited to ModelArchive (https://modelarchive.org/) with the accession code ma-6e3da [https://doi.org/10.5452/ma-6e3da]. Source data are provided with this paper.

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

## Acknowledgements

We thank Chris Richardson and Ruth Knight for support with high-performance computing and protein expression in insect cells, respectively. We thank A. Radzisheuskaya for help with knock-out cell line verification, and we acknowledge S. Jackson and J. Mansfeld for providing the RPE p53-/- FRT/TR cell line and the pcDNA5-Neo-FRT/TR plasmid, respectively. Data for two of the high-resolution cryo-EM structure reported in this paper were collected at the London Cryo-EM (LonCEM) consortium facility, which is supported by Wellcome Grant 206175/Z/17/Z. J.F. was supported by an ICR postdoctoral fellowship. B.J.G. was supported by a Career Development Award from the Medical Research Council of the UK, grant MR/V009354/1. Work conducted in the laboratory of W.N. by P.R.M. and M.L. was supported by MRC research grant MR/X018547/1. For the purpose of Open Access, the authors have applied a CC BY public copyright licence to any Author Accepted Manuscript (AAM) version arising from this submission.

## Author contributions

B.J.G. designed the structure-based project. J.F. cloned, expressed, and purified DNA repair factors for structural studies. J.F. prepared cryo-EM specimens. J.F., N.B.C., and B.J.G. acquired cryo-EM data. T.M.-P. supported data collection on the Glacios cryo-TEM. J.F. processed cryo-EM data, and with support from B.J.G., J.F., and B.J.G. built and refined atomic models. J.F. and P.R.M. constructed knock-out and stable human cell lines. P.R.M. performed assays involving human cells in the lab of W.N., supported by S.K. for co-immunoprecipitations and Western blotting and by M.L. for preparation of constructs and cell handling. All authors contributed to interpretation of the data. B.J.G., J.F., and P.R.M. prepared the figures. B.J.G. wrote the initial draft of the paper with contributions from J.F. and P.R.M., and all authors contributed to its final form.

## Competing interests

The authors declare no competing interests.
