## [Transparent Peer Review file · Nature Communications]

Molecular basis of XPF-ERCC1 targeting to SLX4-dependent DNA repair pathways

Corresponding Author: Dr Basil Greber

Version 0:

Reviewer comments:

Reviewer #1

(Remarks to the Author)

Revisions for Nature Communications

This work by feng and colleagues provides unprecedented structural insight of the XPF-ERCC1/SLX4330-555/SLX4IP and XPF-ERCC1/SLX4IP protein complexes obtained through a combination of cryo-EM studies and biochemical assays. The cryo-EM structure of XPF-ERCC1/XPA complex, which has been previously established by Neil McDonald and colleagues (Jones et al. Nat Comm 2020), is also determined and used for comparison. Importantly, the authors also have determined the cryo-EM structure of the XPF-ERCC1/SLX4330-555/SLX4IP complex bound to a splayed arm Y DNA structure.

A detailed structural view of the binding interfaces of XPF with both SLX4330-555 and SLX4IP is provided. Previous studies had mapped the XPF-binding and SLX4-binding domains of SLX4 and XPF, respectively, and had identified residues critical for XPF-SLX4 complex formation. This study provides additional confirmatory structural insight showing that XPF-SLX4 complex formation relies on a short 526-555 region of SLX4. Importantly, no other part of the SLX4330-555 fragment appears to contact XPF in the apo complex whereas additional contacts spanning residues 467 to 492 and 500 to 512 are seen when the complex is bound to DNA.

The mechanism by which SLX4 stimulates XPF activity remains unclear as the XPF/SLX4 binding interface is found to be remote from the catalytic domain of XPF. The authors speculate that the interaction between the second UBZ domain of SLX4 (UBZ-2) and the ERCC1 (HhH)₂ domain that is predicted by AF2 may prevent the formation of the auto-inhibited conformation of XPF-ERCC1. This auto-inhibited conformation was previously described by the McDonald lab and is also seen here in the apo complex. However, this hypothesis was excluded as an SLX4378-555 fragment lacking the UBZ-2 domain was just as efficient at stimulating XPF-ERCC1 as the SLX4330-555 fragment. The authors propose that the additional contacts detected between SLX4330-555 and XPF in the DNA bound complex may have allosteric effects that contribute to XPF stimulation. Unfortunately, this was not experimentally addressed.

In the second part of the manuscript the authors focus on the SLX4IP-XPF interaction, which can occur in the absence of SLX4. They provide the first structural views of SLX4IP and help better characterize how it interacts with XPF. Interestingly, the cryo-EM structure of XPF-ERCC1/SLX4330-555/SLX4IP shows that XPF is sandwiched between SLX4 and SLX4IP, which bind opposite sides of XPF, and that SLX4IP makes no direct contact with SLX4330-555. To further investigate how XPF and SLX4IP can directly interact, they determined the cryo-EM structure of an XPF-SLX4IP complex without SLX4330-555. Remarkably, the structure of the XPF-SLX4IP complex turns out to be nearly identical with or without SLX4330-555. These observations along with a series of additional data provided by the authors are important as they strongly challenge the view that SLX4IP and XPF share the same binding interface on SLX4. They also have facilitated the generation of new separation-of-function amino acid substitution mutations (notably XPF 544-549 3AK, I554P and E572R) which lay the foundation for future mechanistic dissection experiments. Another interesting observation is that the L16/V17 and V115/V116 residue pairs that are part of putative SIMs in the N-terminus of SLX4IP are buried between SLX4IP and XPF. This suggests that the previously proposed contribution of the SIMs to the telomeric localization of SLX4IP would occur only when it is not bound to XPF.

This is overall an interesting study that provides a set of unprecedented observations on a timely topic. However, while the data appear to be robust and are convincing, this study would benefit from additional experimental work to support some of the hypotheses made by the authors. The structure of the manuscript would also benefit from some remodeling (see below).

Questions/ Comments.

- Panier and colleagues showed that immunoprecipitation of a short GFP-SLX41-200 fragment over-expressed in HEK293 cells can pull down endogenous SLX4IP (Fig S2 1A, Panier et al Mol Cell 2019), even though it lacks the UBZ domains, the XPF-binding MLR domain and the BTB homodimerization domain which could directly or indirectly interact with XPF-ERCC1. This seems incompatible with a situation where SLX4 and SLX4IP never make direct contacts and that their interaction is exclusively bridged by XPF. Maybe this falls outside of the scope of the current study, but it seems worthwhile to determine the structure of an XPF-ERCC1/SLX4/SLX4IP complex that contains a longer fragment of SLX4 than the one used here. This could help identify potential direct contacts between SLX4 and SLX4IP and would help refine our understanding of the functional interactions between XPF-ERCC1, SLX4IP and SLX4.
- Why not test the hypothesis that the additional contacts made by SLX4330-555 with XPF in the DNA-bound complex could be important for catalytic stimulation of XPF-ERCC1? This could be easily tested in vitro by comparing the catalytic activity of XPF-ERCC1 and XPF-ERCC1 in complex with N-terminally truncated versions of the SLX4330-555 fragment. The authors have already shown that an SLX4378-550 fragment is just as efficient at stimulating XPF-ERCC1 as the reference SLX4330-555 fragment. What about shorter SLX4 fragments lacking the motifs that make contact with XPF in the DNA-bound complex?
- Have the authors tried to determine the cryo-EM structure of the XPF-ERCC1/SLX4 complex +/- DNA **without SLX4IP**? This seems like an important addition to this study, especially since SLX4IP is totally dispensable for catalytic stimulation of XPF-ERCC1.
- page 6 lines 26-28: The authors should tame down their description of the results by writing that “the experiments performed using cells transfected with the SLX45A mutant construct showed **strongly impaired** (instead of “did not show an”) interaction with XPF, ERCC1, or SLX4IP...” as faint bands for each one of these proteins are detected in the GFP-SLX45A GFP pull down. This could result from heterodimerization between SLX45A and endogenous SLX4. It would be worthwhile testing this by depleting endogenous SLX4 with siRNAs. If no more XPF-ERCC1 and SLX4IP are detected, then the authors can claim to have generated a null XPF-binding mutant. The absence of SLX4IP would also indicate that should it establish any contact with SLX4 (see first comment above), it can only do so within the context of an SLX4-XPF-ERCC1 complex.
- Is SLX4IP co-immunoprecipitated with SLX4 in cells producing an XPF544-549 3AK mutant that is strongly impaired in SLX4IP binding?
- The manuscript would strongly benefit from the addition of a schematic in Figure 1 showing the full-length proteins with the position and distribution of their functional domains, as well as schematics of the fragments used in the study. This would provide useful support to readers that are not experts on XPF-ERCC1, SLX4 and SLX4IP.
- It would help the flow of the Results section if the rather long discussion around DNA binding within the XPF-ERCC1/SLX4330-555/SLX4IP complex was moved to the Discussion sections.
- Page 9 line 29: No solid evidence seems to be presented that would allow to state “...confirming that XPF is an enzyme employing two-metal-ion catalysis”. ‘Consistent with’ doesn’t equate to ‘confirmed’. The authors should remain speculative in this closing statement, as the data are not definitive
- Page 13 line 27: The UBZ-2 of SLX4 is not required for ICL repair (cf Lachaud et al JCS 2014). The discussion around UBZ-2 needs to be modified accordingly.

Reviewer #2

(Remarks to the Author)

Reviewer #3

(Remarks to the Author)

The manuscript by Feng and colleagues focuses on the structural basis of DNA repair involving the XPF-ERCC1 endonuclease complex, a key player in multiple DNA repair pathways. The authors present the structures of human XPF-ERCC1 in complex with two important DNA repair factors: SLX4 and SLX4IP. These new structures identify critical residues mediating the interactions and provide a structural rationale for how specific mutations disrupt these protein-protein interfaces in human cells. In addition, the study reports the structure of the XPF-ERCC1-SLX4-SLX4IP complex bound to DNA. The structures appear to be of high quality. While this study provides novel data of potential interest to a broad

scientific audience, the manuscript would benefit from substantial revision to improve the clarity of its message and enhance its accessibility to a wider audience.

Major concern :

- 1) On several occasions, the main message of the article is less clear, as it becomes confused by secondary information. For example, Figure 1 presents the AlphaFold2-predicted model, but not the experimental model (determined at 2.9 Å resolution), which is in fact the one that will be deposited in the Protein Data Bank. While AlphaFold2 has undeniably revolutionized structural biology, its role in helping to design the structures used in this study could be more appropriately described in the Methods section. Similarly, technical details such as the types of microscope and camera used for data collection (p. 4 – p. 5) should be moved to the Methods section.
- 2) Page 5, line 23 : The authors report the first structure of the N-terminal region of SLX4IP and should provide a more detailed description of this new structure. The term 'compact fold' is too vague.
- 3) Figure 3b : Was the nuclease activity assay repeated three times ? (this is not indicated in the method section ; for such nuclease activity assays the experiment must be reproduced at least 3 times independently) For clarity, the authors may consider quantifying the intensity of the band corresponding to the uncleaved substrate, in order to compare the variation in nuclease activity across the different XPF complexes. These results could be presented as histogram bars.
- 4) Page 7, lines 11-17 : The authors suggest that residues that were not observed in the structure might be able to transiently or dynamically form interactions with XPF-ERCC1 to increase substrate cleavage. The authors should consider alternative mechanisms, e.g. allosteric effect, ...
- 5) Page 11, lines 2-3 : The absence of a visible interaction between SLX4 and SLX4IP in the structure does not necessarily imply that SLX4IP binding is independent of SLX4. Cooperative binding through an allosteric mechanism remains a possible explanation.
- 6) Figure 8 is informative but should be extended. The schematic model should not only highlight the critical contacts involved in complex assembly, but also illustrate the functional implications of these interactions on the biological activity of the different components.

Minor concerns :

- 1) Page 5, Line 27 : I would suggest replacing « High-resolution » by « 2.9 Å cryo-EM structure of ... »
- 2) Page 8, Line 6 : Consider rephrasing : « the dsDNA-single-stranded DNA (ssDNA) junction »

Reviewer #4

(Remarks to the Author)

Feng et al report novel cryo-EM structures of and biochemical analysis of a series of XPF-ERCC1 complexes with XPA, SLX4-SLX4IP and a five-protein complex of XPF-ERCC1-SLX4-SLX4IP-DNA complexes of ercc1-xpf SLX4-SLX4IP-DNA bound to dsDNA. Overall, several new important observations are gleaned from the study including the molecular basis of SLX4 and SLX4IP engagement of the XPF-ERCC1 heterodimeric nuclease and the molecular basis for human XPF-ERCC1 engagement of dsDNA. The paper is nicely illustrated and well-written. A deficiency in this work stems from the fact that based on the structures, novel mutations are made to probe the role for elements of binding and protein-protein interactions, but these efforts are not extended to functional analysis of the mutants in the DNA damage response.

1. Figure 7. The authors report that structure-based mutations in the SLX4IP- XPF interface disrupt the protein-protein interaction. However, other than binding, there is no functional readouts for these mutants, making the analysis preliminary. Can the authors evaluate the mutations or some subset of them on SLX4IP in vitro or in vivo functions?
2. The structural analysis is initiated with an Alphafold prediction of the mode of binding of SLX4 and SLX4IP binding to XPF-ERCC1. Throughout the paper the authors make comparisons to the Alphafold models. I feel like this distracts from the high-quality reliable experimental structural information reported in the manuscript. It would be my recommendation to focus more on experimentally determined aspects of the complexes, rather than litigate the validity of the models.
3. The presentation of the data statistics in Supplementary Tables 1 and 2 is a little confusing as the authors do not use precise nomenclature for the datasets that matches them to the relevant cryo-EM datasets. For example, the XPF-ERCC1-XPA complex is simply labeled XPA in Supp Table 1. For clarity, please consolidate the nomenclature.
4. The authors describe fitting of DNA duplex in the text and raise the caveat that: “we acknowledge that the DNA might be bound in multiple sequence registers or orientations, and we restrict the detail level of our interpretation accordingly.” However, they appear to have modeled and refined a specific sequence register with 40 nucleotides modeled and refined reported in Supp. Table 2? Please clarify.
5. The authors argue that catalytic metals are bound, but the DNA does not appear to be cleaved. Could the DNA binding mode be reflective of a DNA scanning mode for the complex? Are there indications in the active site geometry that would suggest why the backbone is not cleaved? Perhaps the authors can comment on why the DNA is not cleaved. Is the issue alignment of catalytic metals, the substrate being dsDNA in the active site versus a ssDNA-dsDNA junction, or both.

Version 1:

Reviewer comments:

Reviewer #1

(Remarks to the Author)

We appreciate the work undertaken to address the comments made by ourselves and fellow reviewers. Overall, the

comments were addressed well, with particular care taken to reword potentially misleading sentences. It is also clear that significant effort was made to experimentally reinforce certain areas of the study in line with our suggestions. In particular, we think the additional in vitro cleavage data using N-terminally truncated versions of the SLX4330-555 fragment improves our understanding of the functional effects of the SLX4-XPF interaction, although we would have liked to see more incremental truncations of this SLX4 fragment to further elucidate the contributions of individual motifs in the region. The novel structural insight into the XPF-ERCC1/SLX4330-555/SLX4IP and XPF-ERCC1/SLX4IP protein complexes provided by this study is undoubtedly a valuable resource to all working on the topic. We do not see any reason to delay its publication from our standpoint.

Reviewer #2

(Remarks to the Author)

Reviewer #3

(Remarks to the Author)

reviewer 2: The authors have satisfactorily addressed all my questions. The revised version of their work has significantly improved the manuscript. I can only congratulate the authors on this excellent piece of work.

reviewer 4's comments: Overall, reviewer 2 thinks that the authors have adequately addressed the comments by reviewer 4. Maybe, regarding the point (5) addressed by reviewer 4, the authors could add an alternative hypothesis to the one that was added to the revised manuscript: "The comparably low efficiency of this dsDNA nicking reaction might be why we were able to observe uncleaved DNA substrate in our cryo-EM complexes." The authors may consider adding the alternative hypothesis that the complex is not in a catalytic mode but rather, in a scanning mode.

Feng et al., Nature Communications, Response to Reviewers

Reviewer #1 (Remarks to the Author):

Revisions for Nature Communications

This work by feng and colleagues provides unprecedented structural insight of the XPF-ERCC1/ SLX4330-555/SLX4IP and XPF-ERCC1/SLX4IP protein complexes obtained through a combination of cryo-EM studies and biochemical assays. The cryo-EM structure of XPF-ERCC1/XPA complex, which has been previously established by Neil McDonald and colleagues (Jones et al. Nat Comm 2020), is also determined and used for comparison. Importantly, the authors also have determined the cryo-EM structure of the XPF-ERCC1/ SLX4330-555/SLX4IP complex bound to a splayed arm Y DNA structure.

[...]

This is overall an interesting study that provides a set of unprecedented observations on a timely topic. However, while the data appear to be robust and are convincing, this study would benefit from additional experimental work to support some of the hypotheses made by the authors. The structure of the manuscript would also benefit from some remodeling (see below).

We thank the reviewer for the detailed assessment of our work and for the suggestions provided.

Questions/ Comments.

Panier and colleagues showed that immunoprecipitation of a short GFP-SLX41-200 fragment over-expressed in HEK293 cells can pull down endogenous SLX4IP (Fig S2 1A, Panier et al Mol Cell 2019), even though it lacks the UBZ domains, the XPF-binding MLR domain and the BTB homodimerization domain which could directly or indirectly interact with XPF-ERCC1. This seems incompatible with a situation where SLX4 and SLX4IP never make direct contacts and that their interaction is exclusively bridged by XPF.

We have added the following statement to this section (page 12, line 8) to clarify:

“Our data do not exclude the formation of direct interactions between SLX4IP and other parts of SLX4 that are not involved in XPF-ERCC1 binding, for example within the N-terminal 200 residues of SLX4 (Panier et al., 2019).”

Maybe this falls outside of the scope of the current study, but it seems worthwhile to determine the structure of an XPF-ERCC1/SLX4/SLX4IP complex that contains a longer fragment of SLX4 than the one used here. This could help identify potential direct contacts between SLX4 and SLX4IP and would help refine our understanding of the functional interactions between XPF-ERCC1, SLX4IP and SLX4.

This is an interesting suggestion. However, it is worth considering that the AlphaFold3 prediction of the SLX4IP-SLX4¹⁻²⁰⁰ complex suggests that the predicted interaction interface on SLX4IP completely overlaps with the interface that SLX4IP forms with XPF. Therefore, even a longer SLX4 fragment is unlikely to form these interactions in the context of XPF-ERCC1-containing complexes. Structure determination of SLX4IP alone in complex with SLX4 residues 1-200 would require an entirely different experimental approach and we agree with the reviewer that this would clearly exceed the scope of our study.

Why not test the hypothesis that the additional contacts made by SLX4330-555 with XPF in the DNA-bound complex could be important for catalytic stimulation of XPF-ERCC1? This could be easily tested *in vitro* by comparing the catalytic activity of XPF-ERCC1 and XPF-ERCC1 in complex with N-terminally truncated versions of the SLX4330-555 fragment. The authors have already shown that an SLX4378-550 fragment is just as efficient at stimulating XPF-ERCC1 as the reference SLX4330-555 fragment. What about shorter SLX4 fragments lacking the motifs that make contact with XPF in the DNA-bound complex?

We appreciate this suggestion, and we have therefore performed the corresponding experiment. We find that removal of the segments that bind XPF-ERCC1 in the presence of a DNA substrate, but not in its absence (resulting in SLX4⁵⁰⁰⁻⁵⁵⁵, a truncated version of SLX4³³⁰⁻⁵⁵⁵), strongly reduces the stimulatory effect of SLX4. Because the SLX4 BTB domain was previously suggested to be involved in XPF activation, we have also added constructs that contain this domain. We found that the presence of the BTB domain does not make a difference relative to the equivalent constructs lacking this domain (along with the intervening linker between SLX4 residues 555 and the BTB domain). These results are now presented in Fig. 5 and Supplementary Fig. 7. The Results and Discussion have been modified accordingly (these text segments are too long to conveniently fit into the point-by-point answers; please see pages 10-11 and 15 of the revised manuscript).

Have the authors tried to determine the cryo-EM structure of the XPF-ERCC1/SLX4 complex +/- DNA **without SLX4IP**? This seems like an important addition to this study, especially since SLX4IP is totally dispensable for catalytic stimulation of XPF-ERCC1.

We agree that this would be an interesting extension of our study. We have attempted to determine this structure several times, but we obtained maps showing only XPF-ERCC1 alone, with residual SLX4 density at best, too weak to even perform image classification for the presence of SLX4. This suggests that even though SLX4 and SLX4IP can bind to XPF-ERCC1 independently, as shown by our biochemical experiments (e.g. Supplementary Fig. 2b), the complex containing only SLX4 may fall apart on the cryo-EM grid. This is a common issue in cryo-EM structure determination, and we have not been able to resolve it at this point.

page 6 lines 26-28: The authors should tame down their description of the results by writing that “the experiments performed using cells transfected with the SLX45A mutant construct showed **strongly impaired** (instead of “did not show an”) interaction with XPF, ERCC1, or SLX4IP...” as faint bands for each one of these proteins are detected in the

GFP-SLX4 5A GFP pull down. This could result from heterodimerization between SLX4 5A and endogenous SLX4. It would be worthwhile testing this by depleting endogenous SLX4 with siRNAs. If no more XPF-ERCC1 and SLX4IP are detected, then the authors can claim to have generated a null XPF-binding mutant. The absence of SLX4IP would also indicate that should it establish any contact with SLX4 (see first comment above), it can only do so within the context of an SLX4-XPF-ERCC1 complex.

We appreciate the reviewer's concern. Accordingly, we have changed the text to account more accurately for the experimental observation of a strongly impaired rather than completely abolished (null mutant) interaction (page 5, line 1):

"Identical experiments performed using cells transfected with the SLX4^{5A} mutant construct showed a strongly impaired interaction with XPF, ERCC1, and SLX4IP while the interactions with MUS81-EME1 remained intact (Fig. 2d)."

Is SLX4IP co-immunoprecipitated with SLX4 in cells producing an XPF544-549 3AK mutant that is strongly impaired in SLX4IP binding?

We have produced the XPF knock-out cells and rescued them with XPF^{WT} and XPF^{3AK} mutant. Unfortunately, we were unable to obtain conclusive results for the subsequent co-immunoprecipitation due to technical difficulties. However, we note that we have already confirmed that co-immunoprecipitation of SLX4IP with XPF^{3AK} is reduced while this XPF variant is still able to interact with SLX4 at normal levels (Fig. 7b).

The manuscript would strongly benefit from the addition of a schematic in Figure 1 showing the full-length proteins with the position and distribution of their functional domains, as well as schematics of the fragments used in the study. This would provide useful support to readers that are not experts on XPF-ERCC1, SLX4 and SLX4IP.

We thank the reviewer for this suggestion. We have added this schematic in what is now Figure 1a.

It would help the flow of the Results section if the rather long discussion around DNA binding within the XPF-ERCC1/ SLX4330-555/SLX4IP complex was moved to the Discussion sections.

We appreciate the suggestion to streamline our text. However, we believe that this section would not be suitable for the discussion and prefer for it to remain in the Results section.

Page 9 line 29: No solid evidence seems to be presented that would allow to state "...confirming that XPF is an enzyme employing two-metal-ion catalysis". 'Consistent with' doesn't equate to 'confirmed'. The authors should remain speculative in this closing statement, as the data are not definitive

We agree with the reviewer, and we have reworded this statement:

“Overall, the available data are consistent with the presence of a second metal ion in our cryo-EM structure and with the idea that XPF is an enzyme employing two-metal-ion catalysis.”

Page 13 line 27: The UBZ-2 of SLX4 is not required for ICL repair (cf Lachaud et al JCS 2014). The discussion around UBZ-2 needs to be modified accordingly.

We have removed this portion of the discussion because the UBZ-2 domain has been found not to be important for XPF-ERCC1 stimulation.

Reviewer #2 (Remarks to the Author):

Reviewer #3 (Remarks to the Author):

The manuscript by Feng and colleagues focuses on the structural basis of DNA repair involving the XPF-ERCC1 endonuclease complex, a key player in multiple DNA repair pathways. The authors present the structures of human XPF-ERCC1 in complex with two important DNA repair factors: SLX4 and SLX4IP. These new structures identify critical residues mediating the interactions and provide a structural rationale for how specific mutations disrupt these protein-protein interfaces in human cells. In addition, the study reports the structure of the XPF-ERCC1–SLX4–SLX4IP complex bound to DNA. The structures appear to be of high quality. While this study provides novel data of potential interest to a broad scientific audience, the manuscript would benefit from substantial revision to improve the clarity of its message and enhance its accessibility to a wider audience.

We thank the reviewer for the positive overall assessment of our results and appreciate the suggestions for improvement.

Major concern :

1) On several occasions, the main message of the article is less clear, as it becomes confused by secondary information. For example, Figure 1 presents the AlphaFold2-predicted model, but not the experimental model (determined at 2.9 Å resolution), which is in fact the one that will be deposited in the Protein Data Bank.

We note that Figure 1c, d did show the experimental structures for the interaction sites of XPF with SLX4 and SLX4IP, i.e. the most important regions for our interpretation, along with the cryo-EM maps in Figure 1b. However, we appreciate the reviewer’s suggestion, and we have removed the AlphaFold predictions from this figure (they are still available in Supplementary Figure 1). We have added overview panels showing the entire cryo-EM-derived atomic models in new panels in Figure 1c and d. Combined with other

comments, this suggestion has helped us produce a more informative figure. We appreciate the helpful suggestion.

While AlphaFold2 has undeniably revolutionized structural biology, its role in helping to design the structures used in this study could be more appropriately described in the Methods section. Similarly, technical details such as the types of microscope and camera used for data collection (p. 4 – p. 5) should be moved to the Methods section.

We appreciate the reviewer's concern and have removed the AlphaFold panels from Figure 1 (they are still available in the supplement), and we have removed the information on the microscopes used from the main text (this information was already present in the Methods section).

We have also shortened and removed other sections that discuss AlphaFold-derived structures or hypotheses (in the Results and the Discussion). However, we prefer to retain a brief description of the AlphaFold-based modelling in the main text because this process also resulted in a hypothesis of XPF activation that we are subsequently testing.

2) Page 5, line 23 : The authors report the first structure of the N-terminal region of SLX4IP and should provide a more detailed description of this new structure. The term 'compact fold' is too vague.

We appreciate the need for more in-depth analysis of the SLX4IP structure. We have added a short statement on the structure of SLX4IP, which forms a beta-grasp fold, to page 5 of the paper:

“Our structure shows that the N-terminal 122 residues of SLX4IP form a compact β -grasp fold, a mixed $\alpha+\beta$ -fold that occurs in a diverse array of both enzymes and structural proteins, including ubiquitin⁴³. In SLX4IP, five β -strands wrap around a single long α -helix in a $\beta_2-\alpha-\beta_3$ configuration. The SLX4IP structure resembles the structure of MAJIN, a component of the meiotic telomere complex, which tethers telomere ends to the nuclear envelope during meiosis (Supplementary Fig. 4c-e)^{44,45}.”

Additionally, we have added supplementary figure panels (Supplementary Fig. 4c, d, e) as well as a comparison to the MAJIN subunit of the meiotic telomere complex on page 13.

3) Figure 3b : Was the nuclease activity assay repeated three times ? (this is not indicated in the method section ; for such nuclease activity assays the experiment must be reproduced at least 3 times independently) For clarity, the authors may consider quantifying the intensity of the band corresponding to the uncleaved substrate, in order to compare the variation in nuclease activity across the different XPF complexes. These results could be presented as histogram bars.

We have now provided three replicates of this experiment and the subsequent assay with the SLX4 truncations. These are presented in Figs. 3b and 5c and in Supplementary Figs 5 and 7.

4) Page 7, lines 11-17 : The authors suggest that residues that were not observed in the structure might be able to transiently or dynamically form interactions with XPF-ERCC1 to increase substrate cleavage. The authors should consider alternative mechanisms, e.g. allosterical effect, ...

We appreciate this suggestion, and we have addressed this experimentally. We find that removal of the segments that bind XPF-ERCC1 in the presence of a DNA substrate, but not in its absence (i.e. dynamically and in a state-specific fashion), strongly reduces the stimulatory effect of SLX4. We have also accounted for the possibility of allostery in the text (page 15, first paragraph)

Because the SLX4 BTB domain was previously suggested to be involved in XPF activation as well, we have also added constructs that contain this domain. We found that the presence of the BTB domain does not make a difference relative to the equivalent constructs lacking this domain and the intervening linker between SLX4 residues 555 and the BTB domain. These results are now presented in Fig. 5c.

5) Page 11, lines 2-3 : The absence of a visible interaction between SLX4 and SLX4IP in the structure does not necessarily imply that SLX4IP binding is independent of SLX4. Cooperative binding through an allosteric mechanism remains a possible explanation.

We note that we have purified XPF-ERCC1-SLX4 and XPF-ERCC1-SLX4IP complexes (Supplementary Fig. 2), indicating that both accessory proteins can bind to XPF-ERCC1 independently. However, we agree with the reviewer that binding of one protein could modulate the binding properties of the other.

Additionally, we have added the following statement to this section (page 12, line 6) to clarify that SLX4 and SLX4IP might interact via SLX4 segments that are not present in our sample:

“While our data establish that SLX4 and SLX4IP can bind to XPF-ERCC1 on their own, we cannot exclude that their binding might be synergistic, i.e. that the presence of one binding partner might enhance binding of the other. Our data also do not exclude the formation of direct interactions between SLX4IP and other parts of SLX4 that are not involved in XPF-ERCC1 binding, for example within the N-terminal 200 residues of SLX4 (Panier et al., 2019).”

6) Figure 8 is informative but should be extended. The schematic model should not only highlight the critical contacts involved in complex assembly, but also illustrate the functional implications of these interactions on the biological activity of the different components.

We appreciate that the reviewer considers our summary figure (Fig. 8) useful and worthy of extension. We have added further information that provides a more complete overview of complex assembly and component activation.

Minor concerns :

1) Page 5, Line 27 : I would suggest replacing «High-resolution» by «2.9 Å cryo-EM structure of ... »

We appreciate the suggestion and have made the change as advised.

2) Page 8, Line 6 : Consider rephrasing: «the dsDNA-single-stranded DNA (ssDNA) junction»

We have now introduced the “ssDNA” abbreviation earlier in the text, which has allowed us to replace this text segment with “ssDNA-dsDNA junction”.

Reviewer #4 (Remarks to the Author):

Feng et al report novel cryo-EM structures of and biochemical analysis of a series of XPF-ERCC1 complexes with XPA, SLX4-SLX4IP and a five-protein complex of XPF-ERCC1-SLX4-SLX4IP-DNA complexes of ercc1-xpf SLX4-SLX4IP-DNA bound to dsDNA. Overall, several new important observations are gleaned from the study including the molecular basis of SLX4 and SLX4IP engagement of the XPF-ERCC1 heterodimeric nuclease and the molecular basis for human XPF-ERCC1 engagement of dsDNA. The paper is nicely illustrated and well-written. A deficiency in this work stems from the fact that based on the structures, novel mutations are made to probe the role for elements of binding and protein-protein interactions, but these efforts are not extended to functional analysis of the mutants in the DNA damage response.

We appreciate the importance of exploiting the insights from our study for a functional analysis of XPF-ERCC1 and SLX4IP in the context of SLX4-driven DNA repair activities. We have therefore generated XPF knock-out cells using CRISPR-Cas9 technology and used these for an initial functional characterisation of the XPF mutants that impair binding of SLX4IP (see below). However, a complete functional characterisation of this DNA repair system is beyond the scope of this study.

1. Figure 7. The authors report that structure-based mutations in the SLX4IP- XPF interface disrupt the protein-protein interaction. However, other than binding, there is no functional readouts for these mutants, making the analysis preliminary. Can the authors evaluate the mutations or some subset of them on SLX4IP in vitro or in vivo functions?

We have added a cis-platin challenge survival assay using XPF knock-out cells that were complemented with XPF^{WT} or the XPF^{3AK} mutant (deficient in the SLX4IP interaction). We observe that the cells expressing XPF^{3AK} are substantially more sensitive to cis-platin than XPF^{WT} cells, confirming the functional relevance of this interaction (Fig. 7c). As noted above, a comprehensive analysis of this repair pathway is beyond the scope of our study.

2. The structural analysis is initiated with an Alphafold prediction of the mode of binding of SLX4 and SLX4IP binding to XPF-ERCC1. Throughout the paper the authors make

comparisons to the Alphafold models. I feel like this distracts from the high-quality reliable experimental structural information reported in the manuscript. It would be my recommendation to focus more on experimentally determined aspects of the complexes, rather than litigate the validity of the models.

We thank the reviewer for this assessment. We have reduced the prominence of certain elements depicting the predictions (e.g. in Figure 1, see above), and we have shortened the discussion relating to the SLX4 UBZ-2 domain, in agreement with suggestions from other reviewers. However, we note that the predicted binding of the SLX4 UBZ-2 domain on the ERCC1 HhH₂ domain has direct implications on an important functional question – XPF activation – and can therefore not be ignored completely. Readers interested in this endonuclease system will immediately wonder about this when viewing our predictions (now in the Supplementary information) or running their own predictions. While we therefore believe that this cannot be removed from the manuscript completely, we have now further restricted the discussion of this topic and moved figure panels comparing the structure and the prediction to the supplementary information (also in the light of our new experimental findings presented in Fig. 5c).

3. The presentation of the data statistics in Supplementary Tables 1 and 2 is a little confusing as the authors do not use precise nomenclature for the datasets that matches them to the relevant cryo-EM datasets. For example, the XPF-ERCC1-XPA complex is simply labeled XPA in Supp Table 1. For clarity, please consolidate the nomenclature.

We appreciate that this may cause confusion, and we have added “XPF-ERCC1-“ to all dataset labels to clarify which complex was imaged.

4. The authors describe fitting of DNA duplex in the text and raise the caveat that: “we acknowledge that the DNA might be bound in multiple sequence registers or orientations, and we restrict the detail level of our interpretation accordingly.” However, they appear to have modeled and refined a specific sequence register with 40 nucleotides modeled and refined reported in Supp. Table 2? Please clarify.

The overwhelming majority of the DNA molecules within our cryo-EM particle population is likely to be bound in the refined sequence register, which was derived from the cryo-EM map (see Supplementary Fig. 6g). However, given the absence of direct recognition of a defined secondary structure feature by the endonuclease, it is possible that a small fraction of the particle population contains the DNA molecule in positions that are accessible by XPF-ERCC1 “sliding” up and down on the DNA.

We have thus re-worded this statement: “Even though the purine-pyrimidine pattern establishes the prevalent sequence register of the bound DNA substrate, we acknowledge that, given the absence of ssDNA-dsDNA junction recognition, a small fraction of our cryo-EM particles might contain the DNA bound in a different sequence register.”

This is similar to e.g. amino acid side chains, which may also access more than one rotameric conformation even if one conformation dominates. We hope that this clarifies the intended meaning.

5. The authors argue that catalytic metals are bound, but the DNA does not appear to be cleaved. Could the DNA binding mode be reflective of a DNA scanning mode for the complex? Are there indications in the active site geometry that would suggest why the backbone is not cleaved? Perhaps the authors can comment on why the DNA is not cleaved. Is the issue alignment of catalytic metals, the substrate being dsDNA in the active site versus a ssDNA-dsDNA junction, or both.

The observed dsDNA nicking activity of XPF-ERCC1 is low. The reason for the low efficiency of cleavage, and thus our ability to capture this complex in a cryo-EM structure, is thus most likely the fact that dsDNA is bound. We note that around half of the complexes initially picked from the micrographs do not contain DNA. It is of course impossible to determine with any certainty why this is the case, but one possibility is that these have cleaved the substrate and then dissociated (another possibility is that they never bound DNA).

We have added the following sentence to this section (page 8, line 9) to clarify:

“The comparably low efficiency of this dsDNA nicking reaction might be why we were able to observe uncleaved DNA substrate in our cryo-EM complexes.”

Feng et al., Nature Communications, response to reviewers, round 2

Reviewer #1 (Remarks to the Author):

We appreciate the work undertaken to address the comments made by ourselves and fellow reviewers. Overall, the comments were addressed well, with particular care taken to reword potentially misleading sentences. It is also clear that significant effort was made to experimentally reinforce certain areas of the study in line with our suggestions. In particular, we think the additional in vitro cleavage data using N-terminally truncated versions of the SLX4330-555 fragment improves our understanding of the functional effects of the SLX4-XPF interaction, although we would have liked to see more incremental truncations of this SLX4 fragment to further elucidate the contributions of individual motifs in the region. The novel structural insight into the XPF-ERCC1/SLX4330-555/SLX4IP and XPF-ERCC1/SLX4IP protein complexes provided by this study is undoubtedly a valuable resource to all working on the topic. We do not see any reason to delay its publication from our standpoint.

We thank the reviewer for the positive evaluation of our revised manuscript.

Reviewer #2 (Remarks to the Author):

Reviewer #3 (Remarks to the Author):

reviewer 2: The authors have satisfactorily addressed all my questions. The revised version of their work has significantly improved the manuscript. I can only congratulate the authors on this excellent piece of work.

We appreciate the kind assessment of our revised manuscript.

reviewer 4's comments: Overall, reviewer 2 thinks that the authors have adequately addressed the comments by reviewer 4. Maybe, regarding the point (5) addressed by reviewer 4, the authors could add an alternative hypothesis to the one that was added to the revised manuscript: "The comparably low efficiency of this dsDNA nicking reaction might be why we were able to observe uncleaved DNA substrate in our cryo-EM complexes." The authors may consider adding the alternative hypothesis that the complex is not in a catalytic mode but rather, in a scanning mode.

We have considered this possibility. However, it does not appear to be generally accepted that XPF works by scanning (see e.g. Houtsmuller et al., Science, 1999). We therefore prefer not to add this additional (possibly contentious) hypothesis to this paragraph.